# A Principled Evaluation Framework for Neuron Explanations

## Abstract

Understanding the function of individual units in a neural network is an important building block for mechanistic interpretability. This is often done by generating a simple text explanation of the behavior of individual neurons or units. However, for these explanations to be useful, we must understand how reliable and truthful they are. In this work we unify many existing explanation evaluation methods under one mathematical framework. This allows us to compare and contrast existing evaluation metrics and understand the evaluation pipeline with increased clarity. We propose two simple sanity checks on the evaluation metrics and show that many commonly used metrics fail these tests and do not change their score after massive changes to the concept labels. Based on our experimental and theoretical results, we propose guidelines that future evaluations should follow and identify good evaluation metrics such as correlation.

## 1 Introduction

Deep learning models have achieved great success on a wide range of tasks, but they are very difficult to understand and often perceived as black-boxes. To address this challenge, the field of mechanistic interpretability has recently emerged, aiming to provide a clearer understanding of the internal mechanism of deep neural networks.

Providing natural language explanations for small components of a neural network is an important part of mechanistic interpretability. Classic work in this area includes Network Dissection [1] and other works explaining individual neurons in deep vision models [2; 3; 4; 5; 6; 7; 8; 9]. Other examples include automated neuron explanations [10; 11] for large language models, as well as explaining features of sparse autoencoders [12; 13].

Despite the introduction of various approaches for generating neuron explanations, these methods often use completely different metrics to evaluate how good their descriptions are, and it is not clear how they compare to each other. In addition, many evaluation metrics used have problems, as shown by [14] for example. To ensure that unit explanations are reliable and trustworthy, it is crucial to establish a standardized framework for evaluation.

Motivated by the need for a standardized approach, in this work we unify many existing evaluation methods under a single mathematical framework, which provides much needed conceptual clarity to the topic of explanation evaluation. This framework allows us to clearly compare and contrast of current evaluation techniques and provides a more transparent understanding of the evaluation pipeline. Through the framework, we rigorously analyze and identify several failure modes in commonly-used metrics. Additionally, we introduce two sanity tests to validate the metrics, revealing that most commonly used evaluations fail at least one of these basic tests. This helps understand which metrics are not suitable for reliable interpretation and should not be used.

In summary, in this paper we:

- Formalize the task of evaluating individual unit explanations.
- Unify existing evaluation methods under a single mathematical framework.
- Propose two sanity checks for evaluation metrics: Missing Labels test and Extra labels test, and show that many commonly used metrics fail at least one of these basic tests

- Experimentally compare different evaluation metrics, and identify good metrics to use, such as Correlation.
- Discover that using biased *top-and-random* sampling makes a good evaluation metric such as correlation fail the Extra Labels test, highligting the need for unbiased sampling.

## 2  DEFINITIONS

### 2.1  WHAT IS A INDIVIDUAL UNIT IN A NEURAL NETWORK?

In this paper we are focused on individual unit explanations. By a unit, we mean a smaller part of a neural network that can have an independent meaning. The simplest form of such units is a single neuron, or a single channel of a Convolutional Neural Network (CNN), but a unit can be any scalar function of network inputs. Other interesting units that fit in our framework are linear combinations of neurons (i.e. directions in activation space), which are considered to correspond to a specific interpretable concept. These are used in studies such as TCAV [15], Concept Bottleneck Models [16], Linear Probing [17] or steering vectors [18]. Finally, a unit could be a feature of a Sparse Autoencoder [19; 12] trained to disentangle a layer's activations into interpretable individual components. While other, larger units such as entire layers or attention heads of a model are sometimes of interest, we do not study these in this paper as they have complicated non-scalar activations and require different methods to analyze than our units of interest.

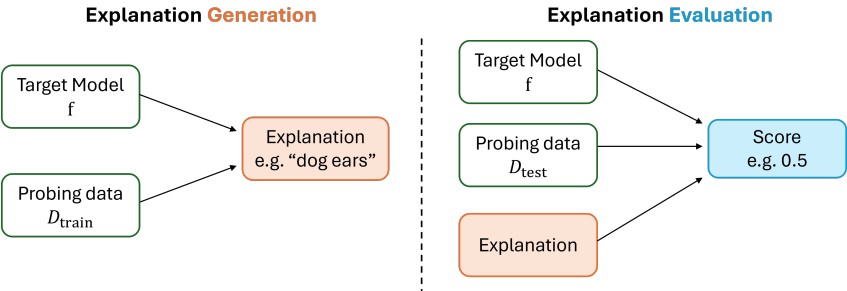

Figure 1: An illustration of the difference between explanation generation and explanation evaluation. Our paper is focused on explanation evaluation.

### 2.2  PROBLEM DESCRIPTION

When it comes to individual unit descriptions, there are two separate, but connected problems: explanation *generation* and explanation *evaluation*, as illustrated in Figure 1. Our focus in this paper is on the *evaluation*, which is a function $\mathcal{E}$ that we formally define below in (2).

**(I) Generation (of Explanation $t_k$):**

$$\mathcal{G}(\mathcal{D}, f, k, l) \rightarrow t_k \tag{1}$$

Here $\mathcal{G}$ is any function, or a process for generating explanations $t_k$ for neuron $k$. This could be a human, an algorithm like Network Dissection [1], or a machine learning model such as MILAN [3]. $\mathcal{D}$ is a probing dataset, $f$ is the network of interest, $k$ is the target neuron and $l$ is the layer of the neuron, and $t_k$ is a text explanation.

**(II) Evaluation (of Explanation $t$):**

$$\mathcal{E}(\mathcal{D}, f, k, l, t) \rightarrow s \tag{2}$$

Here $\mathcal{E}$ is a function that takes a neuron and a text description $t$, and returns a scalar score $s$, where a higher score $s$ indicates the description is better i.e. more reliable/faithful.

**(III) Connection between Evaluation and Generation:**

These two tasks are connected via **argmax generation**, a subset of generation methods that generate the description by finding the concept with the highest evaluation score in the set of concepts $C$, using a specific evaluation method.

$$t_k = \arg \max_{t \in C} \ \mathcal{E}(\mathcal{D}, f, k, l, t) \tag{3}$$

For example, Network Dissection [1] generates an explanation by finding the concept that maximizes the Intersection over Union metric on the Broden dataset.

We can see that evaluations $\mathcal{E}$ can be important in both generation and evaluation of explanations. However, the community has not reached an agreement on which $\mathcal{E}$ to use and when. In fact, currently different papers often use different $\mathcal{E}$ without theoretical justifications. Thus, the focus of our paper is to rigorously investigate what are good evaluation metrics $\mathcal{E}$.

## 2.3 What are the goals of an explanation?

Neuron explanations are typically generated to improve our understanding of the model, which can then help for example improve safety and reliability of the models. However, this is vague and hard to measure in a general way. We believe a more precise definition of the goals of an explanation is essential for thinking clearly about how to evaluate them.

For clarify, it is useful to divide the neural network $f(x)$ into two parts, $f^{0:l}$ and $f^{l+1:L}$, where 0 corresponds to the input layer, $f^{i:j}$ represents the $i$ through $j$'th layers of the neural network $f(x)$, $l$ is the layer of the neuron we are interested in and $L$ is the total number of layers in the network. Here we assume the units are neurons for notational simplicity. Then:

$$f(x) = f^{l+1:L}(f_k^{0:l}(x), f_{\neg k}^{0:l}(x)), \tag{4}$$

where $f_k^{0:l}(x)$ is the activation of neuron of interest $k$ in layer $l$, while $f_{\neg k}^{0:l}(x)$ is the activations of all the other neurons in layer $l$.

Most unit explanations can be seen as a simple interpretable approximation of one (or both) of the following functions:

- **Function 1: Input $\rightarrow$ Unit Activation**
  This corresponds to explaining the function: $x \rightarrow f_k^{0:l}(x)$. A good explanation should be able to describe which inputs cause a high unit activation, and which do not.

- **Function 2: Unit Activation $\rightarrow$ Output**
  This corresponds to explaining the function $z \rightarrow f^{l+1:L}(z, f_{\neg k}^{0:l}(x))$, where $z$ is a real number (e.g. intervened value). This function describes how changes in the unit activation change the final network output.

As we can see from the above notations, these are different problems, and may require different methods to solve and evaluate. While there is often a connection between these two problems, they are often conflated in existing work, and we believe recognizing the difference can improve our understanding. In particular, we should not expect a single text explanation to describe both functions, especially if the neuron is used in an unexpected way. For example, we can imagine a simple image classification model that was trained on biased data, where the *cat* class has only images of black and grey cats, while the class *table* has several images of red cats laying on tables. We could now find a neuron that only activates when there is a red cat in an image, and could be described as a *red cat* neuron for Function 1. However, the activations of this neuron increase the likelihood of the table class, while decreasing the likelihood of the cat class, requiring a different description for its effects (Function 2). This can partially explain findings like [14] finding that the neuron explanations of [10] have little to no effect on an intervention based evaluation, as the intervention based evaluation corresponds to Function 2, while the explanations of [10] were created to explain $f_k^{0:l}(x)$ (Function 1).

In our paper, we focus on evaluating explanations of **Function 1 only**, i.e. $f_k^{0:l}(x)$, as this is more common in existing evaluations and can be applied more generally, for example to explain linear combinations of neurons that don't have a direct effect on the output such as TCAV [15]. While evaluating Function 2 is also important, this requires different methods and is outside the scope of this work.

## 3 A UNIFIED EVALUATION FRAMEWORK

In this section, we propose a principled framework based on the following insight – Almost all existing methods for **evaluation** $\mathcal{E}$ can be formalized as a function of two vectors: neuron activations $a_k$ of neuron $k$ and concept activations $c_t$ of concept $t$, where

- $a_k$ : denotes the activations of neuron $k$ on probing data $x_i \in \mathcal{D}$. I.e. $[a_k]_i = f_k^{0:l}(x_i)$
- $c_t$ : denotes the presence of concept $t$ on the inputs $x_i \in \mathcal{D}$. I.e. $[c_t]_i = P(t|x_i)$.

For notational convenience, we may use $a_{ki}$ to denote $[a_k]_i$ and $c_{ti}$ for $[c_t]_i$ in this paper. Within this framework, the evaluation score $s$ is a function $M(a_k, c_t)$, i.e. $s = \mathcal{E}(\mathcal{D}, f, k, l, t) = M(a_k, c_t)$, where $M$ is the metric chosen to measure similarity between these vectors. Concept activations $c_t$ can be gathered from different sources such as existing labels, pseudo-labels from a model or a crowdsourced human evaluation. The main focus of this paper is analyzing and comparing different choices of metric $M$, and showing how many commonly used metrics have problematic properties.

### 3.1 METRIC DEFINITIONS

**Binarization.** Many similarity metrics used in literature require the inputs to be binary. Since neuron activations, and concept values from some sources are continuous, we need to binarize these vectors. We will denote this with the binarization function $B : \mathbb{R}^n \to \{0, 1\}^n$.

In the below equations we don't explicitly state which binarization function we use, but typically for neuron activations $a_k$ we use $B = \text{top}_\alpha$, where we take top $\alpha$ fraction of activations to be 1, and others to be 0. We formalize this as $\text{top}_\alpha(z)$:

$$[\text{top}_\alpha(z)]_i = \begin{cases} 1 & \text{if } z_i \geq b_\alpha; \\ 0 & \text{otherwise} \end{cases} \tag{5}$$

where $b_\alpha$ satisfies $\sum_{i=1}^n \frac{\mathbf{1}[z_i \geq b_\alpha]}{n} = \alpha$, and $z \in \mathbb{R}^n$. For example, if $\alpha = 0.05$, then $\text{top}_\alpha$ has 1's for inputs with activations in top-5%, and 0 for others. Note $\alpha$ is a hyperparameter needed for all binary similarity functions. We typically select $\alpha$ independently for each metric by finding the value that performs the best on a small validation split of neurons. For concept vectors $c_t$, we usually binarize by simply rounding, denoted as $B = B_r$, where:

$$[B_r]_i = \begin{cases} 1 & \text{if } z_i \geq 0.5 \\ 0 & \text{if } z_i < 0.5 \end{cases} \tag{6}$$

For metrics derived from binary classification, we define the concept value $c_t$ to be the "prediction", and neuron activation $a_k$ as the "ground truth" or observed variable. This corresponds to framing the evaluation as *simulation*, i.e. trying to predict neuron activation based on concept value. Metrics we label as *Inverse* use the opposite framing, i.e. concept value is the ground truth and neuron activation is the prediction. See Appendix A.2 for more discussion on this. Given this, we can express the terms in Confusion matrix (True Positive (TP), False Positive (FP), False Negative (FN) and True Negative (TN)) in terms of the vectors $a_k$ and $c_t$ as:

$$\text{TP} = B(a_k) \cdot B(c_t), \text{ FP} = \overline{B(a_k)} \cdot B(c_t), \text{ FN} = B(a_k) \cdot \overline{B(c_t)}, \text{ TN} = \overline{B(a_k)} \cdot \overline{B(c_t)}$$

where $\overline{B(\cdot)}$ represents element-wise NOT operation on the binary vector (equivalent to $\mathbf{1} - B(\cdot)$) and $\cdot$ is the vector dot product.

Below we express some of the most important and popular evaluation metrics in terms of the vectors $a_k$ and $c_t$, see Appendix C for the definitions of the remaining metrics we evaluated and additional details on these metrics.

**Binary Classification Metrics**

**1. Recall:** Recall, also known as Sensitivity can be intuitively understood as measuring $\mathbb{P}(\text{concept}|\text{neuron active})$.

$$M(a_k, c_t) = \frac{TP}{TP + FN} = \frac{B(a_k) \cdot B(c_t)}{||B(a_k)||_1} \tag{7}$$

**2. Precision:** Precision, also known as Specificity can be intuitively understood as measuring $\mathbb{P}(\text{neuron active}|\text{concept})$.

$$M(a_k, c_t) = \frac{TP}{TP + FP} = \frac{B(a_k) \cdot B(c_t)}{||B(c_t)||_1} \tag{8}$$

**3. IoU:** Intersection over Union, also known as Jaccard Index is a popular metric that measures $\mathbb{P}(\text{concept AND neuron active})/\mathbb{P}(\text{concept OR neuron active})$.

$$M(a_k, c_t) = \frac{TP}{TP + FP + FN} = \frac{B(a_k) \cdot B(c_t)}{||B(a_k)||_1 + ||B(c_t)||_1 - B(a_k) \cdot B(c_t)} \tag{9}$$

**Other Metrics**

**4. AUC:** Area under ROC curve, can be efficiently calculated as:

$$M(a_k, c_t) = \frac{\sum_{i|B(a_k)_i=0} \sum_{j|B(a_k)_j=1} \mathbb{1}[c_{ti} < c_{tj}] + 0.5 \cdot \mathbb{1}[c_{ti} = c_{tj}]}{||B(a_k)||_1 ||1 - B(a_k)||_1} \tag{10}$$

**5. Correlation:** Pearson's correlation coefficient, a very popular metric for measuring similarity between real valued variables.

$$M(a_k, c_t) = \frac{1}{n} \frac{(a_k - \mu(a_k)) \cdot (c_t - \mu(c_t))}{\sigma(a_k)\sigma(c_t)} \tag{11}$$

Here $n$ is the length of $a_k$ and $c_t$, $\mu$ calculates the mean of the vector and $\sigma$ its standard deviation.

## 3.2 EXISTING WORK AS SPECIAL CASES

As summarized in Table 1, we show how existing evaluation work fits into our evaluation framework. We note that that existing evaluation methods differ from each other on four key ways:

(i) **Evaluation metric** $M$: This is the main focus of our paper, to analyze which evaluation metrics are good choices.

(ii) **How is the concept vector** $c_t$ **determined?** There are many choices for the concept vector $c_t$. These include, but are not limited to: labels from a labeled dataset, using a model to create pseudo-labels, using a human evaluator, or generating new inputs and using the prompts as labels.

(iii) **What is the granularity of the vectors?** The simplest case is full input level activations, i.e. $|\mathcal{D}| = |a_k| = |c_t| = n$. These can also be more specific, for example pixel-level as is the case in Network Dissection, or token level as is often the case for language model explanations.

(iv) **What is the probing dataset** $\mathcal{D}$ **used?** This is an important choice and has a significant effect on the outputs. Typical choices include the training/validation data the model was trained on, a special labeled dataset designed for probing, or different datasets for different concepts. Importantly, the dataset used for evaluation should be disjoint from the dataset used for explanation generation to avoid overfitting.

## 4 SANITY CHECKS FOR EVALUATION METRICS

In this section, we start by analytically demonstrating that Precision and Recall metrics have clear and important failure modes and provide an illustrating failure example in Figure 2. In Sec 4.2 we propose two tests or sanity checks for evaluation metrics to further reveal which metrics discussed in Sec 3.1 are unreliable and in Sec 4.3 we discuss the results of these tests.

| Metric $M$ | Study | Concept Source $c_t$ | Granularity | Domain |
|---|---|---|---|---|
| $\sim$Recall | Highly Activating Inputs Human Eval [1; 4; 20; 5] | Crowdsourced | Whole Input | Vision |
| F1-score | Observation based [14] Sparse probes [17] | Generative + Model Labeled data | Whole Input Per-token | Language Language |
| IoU | IoU on Broden [1; 2; 9] | Labeled data | Per-pixel | Vision |
| Accuracy | CBM - concept Error [16] | Labeled data | Whole Input | Vision |
| $\sim$AUC | Comparative Human Study [21] | Crowdsourced | Whole Input | Vision |
| Inverse AUC | INVERT [8] CoSy AUC [22] | Labeled data Generative | Whole Input Whole Input | Vision Vision |
| Correlation* | Simulation - Correlation Scoring [10] | Model | Per-token | Language |
| Correlation | Simulation - Correlation Scoring [7] | Model | Whole Input | Vision |
| Spearman Correlation* | SAE Automated Interpretability [12; 13] | Model | Per-token | Language |
| Cosine cubed | LF-CBM - Automated [20] | Model | Whole Input | Vision |
| $\sim$WPMI | CLIP-Dissect - Similarity Score [4] | Model | Whole Input | Vision |
| MAD $\sim$MAD | CoSy MAD [22] MAIA [6] Explanation Score [23] | Generative Generative Generative | Whole Input Whole Input Whole Input | Vision Vision Language |

Table 1: Summary table comparing evaluations used in esisting work. $\sim$ indicates using a metric with small differences from our definition, while $*$ indicates use of biased *top-and-random* sampling to evaluate the metric with fewer samples. See Table 10 for an extended version of this table.

### 4.1 IDENTIFYING FAILURE CASES

**(I) Failure Case of Recall.** Recall corresponds to looking at the inputs that activate a neuron the highest, and measuring what fraction of them contain concept $c_t$. This is essentially how most crowd-sourced evaluations are currently done. It is known that only measuring recall could be problematic in binary classification as it ignores performance on negative inputs. In our case, using this metric for evaluation will favor explanations with more generic concepts. As an extreme example, consider a very generic description where the concept $c_t = \mathbf{1}$, where $\mathbf{1} \in \mathbb{R}^n$ is a vector of all ones. This could be a concept like "*image*" or "*entity*" which can be a valid description for almost all images. Plugging these into equation (7), we get:

$$M(a_k, c_t) = M(a_k, \mathbf{1}) = \frac{B(a_k) \cdot \mathbf{1}}{||B(a_k)||_1} = \frac{||B(a_k)||_1}{||B(a_k)||_1} = 1, \tag{12}$$

since $B(a_k)_i \geq 0$, $\forall i$. We see that with the maximally generic concept $c_t = \mathbf{1}$, precision is always 1 regardless of the neuron. This is clearly not desirable or a helpful explanation for understanding the neuron. In short, **Recall is biased towards generic concepts** (large $||c_t||_1$).

**(II) Failure case of Precision.** Measuring only precision has the opposite problem, where it favors concepts that are too specific. Imagine an extremely specific concept, that only activates on one image on the entire dataset. We can write this as $c_t = e_i$, where $e_i$ is a unit vector with 1 on the $i$-th element and 0's elsewhere. The precision of this concept on neuron $k$ is then:

$$M(a_k, c_t) = M(a_k, e_i) = \frac{B(a_k) \cdot e_i}{||e_i||_1} = B(a_k)_i \tag{13}$$

If the neuron activates on this input ($B(a_k)_i = 1$), the concept always reaches maximum precision of 1, regardless of how the neuron activates on other inputs. This is undesirable as we should be explaining all activations of a neuron, not just a small fraction of them. For example, explaining

our hypothetical neuron in Figure 2 as *white cat sitting on a couch* would still achieve maximum precision. In short, **Precision is biased towards specific concepts** (small $||c_t||_1$).

**Highly activating images:**

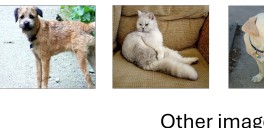 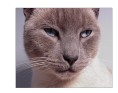 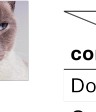

**Other images:**

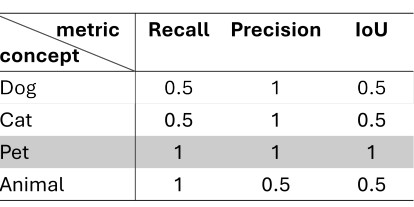

| concept \ metric | Recall | Precision | IoU |
|---|---|---|---|
| Dog | 0.5 | 1 | 0.5 |
| Cat | 0.5 | 1 | 0.5 |
| Pet | 1 | 1 | 1 |
| Animal | 1 | 0.5 | 0.5 |

Figure 2: A hypothetical neuron that only activates on pets (dogs or cats). When comparing different evaluation metrics, we can see recall cannot distinguish between the correct concept (Pet) and a concept that is too generic (Animal), while precision favors concepts that are too specific (Dog, Cat). IoU can unambiguously determine the correct concept.

## 4.2 SANITY TEST DEFINITIONS

Inspired by our above analysis, we propose two general tests to measure whether a certain metric is too biased towards generic or specific concepts.

**Test (I): Missing Labels.** In the missing labels test, we replace $c_t$ with $c_t^-$, where we randomly replace half of the elements of $c_t$ with 0, i.e. we remove half of the concept labels for concept $t$. $\mathbb{E}[||c_t^-||_1] = ||c_t||_1/2$:

$$[c_t^-]_i = \begin{cases} [c_t]_i & \text{with probability } 0.5 \\ 0 & \text{with probability } 0.5 \end{cases} \tag{14}$$

The assumption behind this test is that if concept $t$ is a good description for neuron $k$, removing half of the labels should decrease its similarity score. However, this does not happen with metrics such as Precision that are too biased towards specific concepts, as can be seen in Table 2, causing them to fail this test.

**Test (II): Extra Labels.** Essentially this is the opposite of missing labels test, in which we create $c_t^+$ by randomly doubling the size of $c_t$, i.e. $\mathbb{E}[(||c_t^+||_1)] = 2||c_t||_1$. That is

$$[c_t^+]_i = \begin{cases} 1 & \text{if } [c_t]_i = 1, \text{ else with probability } \frac{||c_t||_1}{n - ||c_t||_1} \\ 0 & \text{otherwise,} \end{cases} \tag{15}$$

where $n$ is the length of vector $c_t$. If concept $t$ is a good description for neuron $k$, giving positive concept labels to random inputs should decrease its similarity score. But this is not the case for methods that are too biased towards generic concepts – i.e. we expect the evaluation metrics such as Recall to fail this test, which is validated in our Table 2. For simplicity we only apply these tests with ground truth labels where $c_t$ is binary.

To perform the tests, we measure two metrics as follows:

$$\text{Score Diff} = \frac{1}{|K|} \sum_{k \in K} M(a_k, c_{t_k}^{\pm}) - M(a_k, c_{t_k}) \tag{16}$$

$$\text{Decrease Acc} = \frac{1}{|K|} \sum_{k \in K} \mathbb{1}[M(a_k, c_{t_k}^{\pm}) - M(a_k, c_{t_k}) < 0] \tag{17}$$

Here $K$ is the set of neurons looked at and $t_k$ is the best/correct concept for neuron $k$. Note that for score diff we normalized the scores such that maximum of $M$ is 1 and minimum value is 0 to allow for equal comparison between metrics.

| | Missing Labels | | Extra Labels | | Pass | |
|---|---|---|---|---|---|---|
| | Score Diff | Decrease acc | Score Diff | Decrease acc | Exp. | Theor. |
| **Recall** | -0.3378 | 99.00% | 0.0032 | 0.00% | ✗ | ✗ |
| **Precision** | 0.0004 | 49.10% | -0.2160 | 99.80% | ✗ | ✗ |
| **F1-score** | -0.1273 | 93.33% | -0.1333 | 99.80% | ✓ | ✓ |
| **IoU** | -0.1216 | 93.33% | -0.1262 | 99.80% | ✓ | ✓ |
| **Accuracy** | -0.0268 | 57.09% | -0.0242 | 99.05% | ✗ | ✗ |
| **Balanced Accuracy** | -0.1530 | 98.96% | -0.0308 | 89.53% | ✗ | ✗ |
| **Inverse Balanced Acc.** | -0.0128 | 66.86% | -0.0961 | 99.99% | ✗ | ✗ |
| **AUC** | -0.1481 | 95.07% | -0.0304 | 75.31% | ✗ | ✗ |
| **Inverse AUC** | -0.0195 | 75.65% | -0.2440 | 99.99% | ✗ | ✗ |
| **Correlation** | -0.0902 | 99.94% | -0.1010 | 99.99% | ✓ | ✓ |
| **Correlation (top-and-random)** | -0.1274 | 93.35% | -0.0405 | 65.41% | ✗ | ✗ |
| **Spearman Correlation** | -0.0303 | 79.26% | -0.0222 | 64.16% | ✗ | ✗ |
| **Spearman Correlation (top-and-random)** | -0.1083 | 85.31% | -0.0384 | 63.18% | ✗ | ✗ |
| **Cosine** | -0.0851 | 99.95% | -0.0704 | 99.27% | ✓ | ✓ |
| **Cosine cubed** | -0.0891 | 99.67% | -0.1016 | 99.94% | ✓ | ✓ |
| **WPMI** | -0.2606 | 96.48% | -0.0287 | 81.76% | ✗ | ✓ |
| **MAD** | -0.0165 | 65.24% | -0.1755 | 99.17% | ✗ | ✗ |
| **AUPRC** | -0.1210 | 97.17% | -0.1306 | 99.80% | ✓ | ✓ |
| **Inverse AUPRC** | -0.2764 | 99.65% | -0.1927 | 94.31% | ✓ | ✗ |

Table 2: Combined experimental results of our missing labels and extra labels test. We can see most evaluation metrics fail at least one of the tests. In the last report we report whether the metric passed our theoretical missing and extra labels tests, showing close alignment with our experimental results.

## 4.3 TEST RESULTS

We experimentally evaluated these metrics on neurons from 6 different settings, covering final layer neurons, hidden layer neurons, CBM neurons and linear probe outputs on 3 image datasets: Imagenet, Places365 and CUB200. See Appendix G for detailed description of the evaluation setting and results on individual datasets.

In Table 2 we report the averaged results of this test across these two sets of neurons for all different evaluation metrics. For simplicity, we say a metric passes the test if its Score Diff is $< -0.05$ and Decrease Acc $> 90\%$. In Table 2 we mark the methods that fail a test in terms of both Score Diff and Accuracy in red color, while methods that only fail one of these are colored orange.

In addition, we run a *theoretical* version of the Missing/Extra labels test on hypothetical neurons whose activations perfectly match a concept, which we discuss in detail in Appendix B. We find that our theoretical results closely match our empirical observations, and that failure in these tests is closely associated with concept imbalance, with failing metrics performing particularly poorly with imbalanced data where concepts are only rarely positive.

Surprisingly, we can see that most metrics fail at least one of these tests:

- Accuracy and Spearman Correlation perform poorly in both directions as their score is largely determined by the majority of inputs that neither activate the neuron nor have the concept.

- Along with Recall, Balanced accuracy and AUC fail the Extra Labels test and are biased towards generic concepts.

- Precision, Inverse Balanced Accuracy, Inverse AUC and MAD fail the Missing Labels Test and are biased towards specific concepts.

The only methods that pass both tests are F1-Score/IoU, Correlation, Cosine, Cosine Cubed and AUPRC. Since a good evaluation metric should be able to pass these tests, we believe metrics beside these should not be relied on by themselves when evaluating unit explanations. This aligns with our interpretation is caused by poor handling of imbalanced data, as metrics known to work worse for

imbalanced data like accuracy and AUC fail the tests, and metrics designed for imbalanced data like F1-score and AUPRC pass the tests. Finally we analytically studied the expected change in scores for different binary metrics under missing and extra labels conditions in Appendix D, where our results largely agree with the empirical findings.

**Top-and-random sampling:** In addition to metrics described in Section 3.1, we also tested a version of correlation and Spearman correlation that uses top-and-random sampling, where the inputs are evaluated on a random sample of consisting of 50% highly activating inputs and 50% randomly sampled inputs as done by [10; 12]. This is in contrast to the default setting of evaluating on the entire dataset or a uniform random sample. For our top-and-random experiments we sampled 25 inputs from the top 0.2% highest activating inputs and 25 random inputs. We can see that while correlation passes both tests when evaluated on full data, top-and-random sampling makes it fail the extra labels test. This makes sense, as greatly oversampling highly activating inputs makes the method more similar to Recall that only focuses only highly activating inputs. This also explains why [14] found explanations from [10] with very high correlation(top-and-random) scores to have relatively low F1-scores.

**Generative $c_t$:** Interestingly, when using generative models for $c_t$ it may sometimes be necessary to use methods that *fail* the Missing Labels test, as the generative labels themselves are missing labels. See Appendix A.4 for further discussion on this phenomenon.

## 5 EXPERIMENTAL COMPARISON OF SIMILARITY FUNCTIONS

Finally, we directly test how good different metrics are at evaluating explanations for final layer neurons, where we know the *ground truth* concept for that neuron.

### 5.1 ARGMAX GENERATION ON FINAL LAYER

Our first way to meta-evaluate the quality of evaluation metrics is via argmax generation on final layer neurons. Here, the neuron we are explaining is a final layer neuron (after softmax), which has a ground truth explanation corresponding to a single classname in the dataset. We denote this ground truth concept as $t_k^*$. We then measure accuracy defined as:

$$\text{Acc}(M) = \frac{1}{|K|} \sum_{k \in K} \mathbf{1}[\arg \max_{t \in C}(M(a_k, c_t)) = t_k^*] \tag{18}$$

Here $K$ is the set of neurons we evaluate and $C$ is the set of concepts we consider. The idea behind this test is that a good evaluation metric $M$ should give the highest score to the correct concept and therefore high accuracy.

### 5.2 AUC ON FINAL LAYER

Our second way to test is also based on final layer neurons, but testing evaluation directly. We use AUC to measure whether the metric separates the scores of all correct (neuron, explanation) pairs from the scores for incorrect (neuron, explanation) pairs, with high AUC indicating a good metric.

$$AUC(M) = \frac{\sum_{k \in K, t \in C | t \neq t_k^*} \sum_l \mathbb{1}[M(a_k, c_t) < M(a_l, c_{t_l^*})]}{(|C| - 1) \cdot |K| \cdot |K|} \tag{19}$$

Here $n$ is the number of neurons, and $C$ is the concept set used to generate incorrect and correct explanations.

**Experimental Setup.** Similar to section 4, we ran these test on 8 different setups, consisting of 4 separate models, 3 datasets and both gt labels and pseudo-labels as concept source for $c_t$. See Appendix G for detailed description of experimental setup and details on individual models.

For all experiments we split a random 5% of the neurons into validation set. For metrics that require hyperparameters such as $\alpha$, we use the hyperparameters that performed the best on the validation split for each setting. We then report performance on the remaining 95% of neurons. In table 3 we report the average scores and average ranks (i.e. the best metric for each setup gets 1, the worst gets 16) of the metrics across the four setups for both tasks.

| | Average across settings | | | | |
|---|---|---|---|---|---|
| **Metric** | **Argmax acc** | **Argmax Rank** | **AUC** | **AUC Rank** | **Total Rank** |
| Recall | 63.92% | 12.00 | 0.98081 | 13.00 | 12.50 |
| Precision | 80.82% | 10.38 | 0.98252 | 11.25 | 10.81 |
| F1-score/IoU | 84.46% | 8.13 | 0.98073 | 12.00 | 10.06 |
| Accuracy | 81.75% | 10.25 | 0.98023 | 11.50 | 10.88 |
| Balanced Accuracy | 83.61% | 8.25 | 0.99161 | 8.63 | 8.44 |
| Inverse Balanced Acc. | 81.62% | 9.38 | 0.99367 | 8.13 | 8.75 |
| AUC | 82.91% | 9.63 | 0.99196 | 10.88 | 10.25 |
| Inverse AUC | 84.79% | 11.00 | 0.98733 | 9.13 | 10.06 |
| Correlation | **93.83%** | **1.38** | 0.99735 | **2.25** | **1.81** |
| Correlation (top-and-random) | 79.03% | 11.50 | 0.99370 | 7.75 | 9.63 |
| Spearman Correlation | 18.58% | 17.25 | 0.83791 | 17.75 | 17.50 |
| Spearman Correlation (top-and-random) | 54.08% | 16.38 | 0.98381 | 13.75 | 15.06 |
| Cosine | 93.13% | 1.75 | **0.99748** | 2.88 | 2.31 |
| Cosine cubed | 93.43% | 2.88 | 0.99673 | 3.50 | 3.19 |
| WPMI | 83.95% | 7.50 | 0.99484 | 7.00 | 7.25 |
| MAD | 86.16% | 5.63 | 0.98694 | 9.13 | 7.38 |
| AUPRC | 85.19% | 6.50 | 0.99266 | 7.00 | 6.75 |
| Inverse AUPRC | 60.45% | 9.63 | 0.98127 | 10.50 | 10.06 |

Table 3: Comparison of different evaluation metrics, averaged across 8 settings. Lower rank means better performance. Best performing metric in **bold**, and second best underlined for each metric. Overall we can see Correlation, Cosine and Cosine cubed perform the best.

We can see that:

1. In general the metrics that passed our tests in Section 4 perform better than those that didn't.

2. Continuous metrics generally perform better than binary ones. This is likely because binarizing neuron activations loses valuable information, and it is hard to find one binarization threshold $\alpha$ that works well for all neurons, i.e. for both single- and super-class neurons.

3. Overall the best performing metrics were correlation and related methods cosine and cosine cubed. Spearman correlation performed clearly the worst.

## 6 CONCLUSIONS

In this paper, we have created a unified mathematical framework for different evaluation metrics and clarified the definitions of around evaluating unit explanations. We have also performed several sanity tests and experiments to answer the following question:

**Which Evaluation Metric should you use to evaluate explanations?** Considering all the evidence from our experiments study, we lean towards **Correlation** (with uniform sampling) as the best overall metric for evaluating neuron descriptions. While cosine similarity performed similarly in Table 3, unlike other metrics its outputs depend on the mean of neuron activations, which can cause significant errors when explaining neurons whose average activation is different from zero as we show in Appendix F.2. Other good metrics include cosine cubed and AUPRC. F1-score or IoU can also be a good choice, but requires choosing an activation cutoff $\alpha$ and it is unclear how to best make this choice. Our **most important recommendation** is that evaluations should not rely on a single metric that doesn't pass our missing or extra labels test. Combinations of such metrics can still be useful, and for example F1-score could be efficiently evaluated by combining a crowdsourced Recall evaluation with evaluating Precision using a generative model.

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

# A ADDITIONAL DISCUSSION

## A.1 LIMITATIONS

**Framework Limitations:**

First, not every evaluation of neuron descriptions can fit into our framework. Below we split these into few separate cases and discuss whether each case represents a limitation of the framework or not:

**Evaluating Multiple Inputs at once:** Our evaluation framework assumes that the presence of a concept is estimated separately for each input. Many human study based evaluations (e.g. [1], [4]) instead evaluate a group of inputs at once, asking questions like "How well does *concept* match this group of images?". However we believe this is simply a less precise/less objective way of asking whether the concept matches each input separately and does not in general represent a significant limitation for the framework.

**Comparing similarity to "correct" explanation:** Another approach to evaluate neuron descriptions is to compare how close they are to a "correct" description, typically in a text-embedding space. For example, this is the main evaluation used by MILAN [3], where they generate "correct" explanation by asking Mechanical Turk workers to describe neurons based on their most highly activating inputs. We do not think this a very reliable way to evaluate explanations, because it relies on the assumption that there exists a single "correct" text-based explanation for each neuron (and that we have some way of finding it), and we do not think this is the case for many real neurons because of issues like polysemanticity and non-verbal concepts like specific graphical patterns. For these hard-to-interpret neurons it is better to just measure how well our explanation matches the neuron like the metrics in our framework do.

**Non-text based explanations:** While we focus on text based concepts $t$ in our paper, the framework works on non-textual concepts just as well, as long as we have some way of generating a concept vector $c_t$ for that concept. For example, the evaluation of [21] uses a group of highly activating inputs as the concept, and then asks workers whether a new input is similar to those inputs or not. Despite this difference, it can be described neatly within our framework.

**Function 2: Activation → Output:** Probably the most significant limitations of our framework is that it is focused on evaluating function 1 only (Input → Unit Activation) as discussed in Section 2.3, and we believe measure Function 2 is equally important. While currently our framework is meant for function 1 only, we believe many of the ideas and metrics we discussed could be useful in evaluating function 2. For example, in a generative model we could use the same metrics to measure similarity between unit activation and the presence of a specific concept in the output. However in function 2 there are additional considerations and thing like measuring difference in outputs when changing the unit activation are likely more important. We believe extending this framework or creating a similar one for function 2 evaluations is an important direction for future work.

**Experimental Result Limitations:**

Overall we are quite confident in the generality of our results on the missing/extra labels test (Table 2) as they are consistent across final/hidden layer neurons and different datasets, and more importantly we showed theoretically that they are caused by poor metric performance on imbalanced data. Importantly this theoretical result is independent of the data domain, type of concepts or the type of unit in question.

However, it is important to note that passing these sanity checks does not guarantee that the metric is a good metric, but failing them does indicate a metric should not be relied on. This is similar to sanity checks proposed by [24], which have been quite influential in the field of saliency maps/input importance estimation.

On the other hand, our comparison results in Table 3 are mostly focused on final layer neurons or other units where we have ground truth available such a concept neurons inside a CBM. While they consistently prefer certain metrics, it is possible that these final layer neurons are systematically different from other units we are interested in such as hidden layer neurons, and these results should not be relied on too strongly.

## A.2 SIMULATION VS CLASSIFICATION

**Simulation:** In section 3.1 we define the neuron activation $a_k$ as the observed *ground truth* variable, and $c_t$ as the predicted variable. This corresponds to seeing the explanation from a **simulation** point of view, i.e. our goal is to predict how the neuron activates, based on the neurons explanation and current inputs. This gives us binary classification metric definitions that are aligned with those of [14].

**Classification:** However this is an arbitrary choice, and we could just as well define $c_t$ as the observed *ground truth* variable and and $a_k$ as the prediction. This corresponds to a classification view, where our goal is to use neuron $k$ as a classifier for concept $c_t$. In terms of metrics, this doesn't change the definitions for True Positive (TP) or True Negative (TN), but it switches the places of False Positive and False Negative, i.e.

$$\text{TP}_{cls} = B(a_k) \cdot B(c_t) = \text{TP}_{sim}$$

$$\text{FP}_{cls} = B(a_k) \cdot \overline{B(c_t)} = \text{FN}_{sim}$$

$$\text{FN}_{cls} = \overline{B(a_k)} \cdot B(c_t) = \text{FP}_{sim}$$

$$\text{TN}_{cls} = \overline{B(a_k)} \cdot \overline{B(c_t)} = \text{TN}_{sim}$$

This change in framing also affects for metrics which are not symmetric in terms of False Positives and False Negatives. For example, Recall(simulation) = Precision(classification), and Precision(simulation) = Recall(classification).

$$\text{Recall(simulation)} = \frac{\text{TP}_{sim}}{\text{TP}_{sim} + \text{FN}_{sim}} = \frac{\text{TP}_{cls}}{\text{TP}_{cls} + \text{FP}_{cls}} = \text{Precision(classification)}$$

The other binary metric that is sensitive to this framing is balanced accuracy. The metric we called Balanced Accuracy in Section 3.1 corresponds to the simulation version of Balanced Accuracy, so for completeness sake we also included Inverse Balanced Accuracy, which is the classification version.

Finally, the AUC we define in Section 3.1 is AUC(simulation), while we also include AUC in the classification framing we called Inv AUC.

Some related works use the simulation framing, while others use the classification definition, which sometimes causes conflicting definitions of metrics like Recall.

## A.3 EQUIVALENCES

During our analysis we also notice that certain separate metrics are equivalent or very similar to each other.

**Correlation and Cosine similarity:** Calculating correlation between two vectors is equal to normalizing each vector to have mean 0 and then taking their cosine similarity, as shown below:

Let $\hat{x} = x - \mu(x)$ for any vector $x$. Then

$$\text{Cosine}(\hat{a}_k, \hat{c}_t) = \frac{\hat{a}_k \cdot \hat{c}_t}{||\hat{a}_k||_2 ||\hat{c}_t||_2} = \frac{(a_k - \mu(a_k)) \cdot (c_t - \mu(c_t))}{\sqrt{n}\sigma(a_k)\sqrt{n}\sigma(c_t)} =$$

$$\frac{1}{n} \frac{(a_k - \mu(a_k)) \cdot (c_t - \mu(c_t))}{\sigma(a_k)\sigma(c_t)} = \text{Correlation}(a_k, c_t) \tag{20}$$

This explains why the two perform very similarly in our evaluations.

**IoU and F1-score:** Below we show that IoU and F1-score are very closely related. In fact, F1-score can be written as a monotonously increasing function of IoU. This means that for any vectors $x_1, y_1, x_2, y_2$, $\text{IoU}(x_1, y_1) < \text{IoU}(x_2, y_2) \rightarrow \text{F1}(x_1, y_1) < \text{F1}(x_2, y_2)$, so for the purposes of comparing similarites they behave identically, and the choice of which one to use doesn't matter. As their performance was exactly the same in all tasks, we report them in the same row in Table 3.

Intersection over Union (IoU) also known as Jaccard index is defined as

$$\text{IoU} = \frac{TP}{TP + FP + FN} \tag{21}$$

while F1-score also known as Dice-score is defined as:

$$\text{F1} = \frac{2TP}{2TP + FP + FN} \tag{22}$$

Now

$$F1 = \frac{2TP}{2TP + FP + FN} = \frac{2TP}{TP + FP + FN} \cdot \frac{TP + FP + FN}{2TP + FP + FN} \tag{23}$$

$$= 2\text{IoU} \cdot \left( \frac{2TP + FP + FN}{TP + FP + FN} \right)^{-1} = \frac{2 \cdot \text{IoU}}{\text{IoU} + 1} \tag{24}$$

Which is monotonously increasing for $0 \leq \text{IoU} \leq 1$. So using either metric gives the same comparative results.

## A.4 GENERATIVE MODELS AND MISSING LABELS TEST

$c_t$ **from Generative Models:** Evaluation methods that use generative models to generate new data and the concept vector $c_t$ actually naturally have missing labels similar to our missing labels test. This is because the generated inputs serve as positive labels for $c_t$, but the negative inputs are often taken to just be all the existing inputs, even though some of actually do have the concept $t$. So the $c_t$ we get from generative models is actually randomly missing a potentially large portion of the labels. Because of this, most evaluation methods using generative $c_t$ in tabel 1 use methods that fail the missing labels test such as Inverse AUC and MAD. In fact, this is desirable as the missing labels in $c_t$ do not affect the evaluation score for these metrics. However, these metrics alone should not be relied for evaluation score, and instead the best way to use generative models in evaluation is to combine them with a another evaluation that doesn't fail the missing labels test, as is done by [14] who measure Precision using generated data, and measure Recall on existing data with model based pseudo-labels and combine these results into F1-score.

# B  THEORETICAL MISSING/EXTRA LABELS TEST AND CONCEPT INBALANCE

In this section we analyze the effects of missing/extra labels test on a simpler toy setting, where neuron $k$'s activations $a_k$ perfectly match the concept labels of concept $t_k$, i.e. $a_k = c_{t_k}$. In this setting, we assume binary neuron and concept activations, i.e. the neuron's activation is 1 if concept is present on the input, and 0 if its not. Consequently, we do not need to perform additional binarization of concept activations with top $\alpha$ like we did in previous sections.

In this simplified setting, our missing and extra labels tests correspond to being able to differentiate between three concepts as defined in section 4:

1. $c_{t_k}$: The perfect predictor for neuron $a_k$, with precision=1 and recall=1.

2. $c_{t_k}^-$: (Missing labels) This concept has Precision=1 since whenever a concept is present, the neuron is also active, and Recall of 0.5 since only half to inputs where the neuron activate now have the concept.

3. $c_{t_k}^+$: (Extra labels) Inverse from above, this concept has Precision=0.5 and Recall=1.

We then measure whether a metric passes the test by measuring the score difference across:

- Missing labels test: $M(a_k, c_{t_k}^-) - M(a_k, c_{t_k}) < -\epsilon$
- Extra labels test: $M(a_k, c_{t_k}^+) - M(a_k, c_{t_k}) < -\epsilon$

, where $\epsilon$ is a small positive number (0.01 in our case). A good metric should be able to reliably differentiate between these concepts. Interestingly we find that the ability of most metrics to differentiate greatly depends on whether the data is balanced or not.

Since the neuron activations perfectly align with concept $t$ and are binary, the only parameter that can effect the results of our missing and extra labels test is the activation frequency of concept $t$, i.e. what fraction of inputs $x \in \mathcal{D}$ contain concept $t$. Following the notation in section D, we denote this fraction as $\gamma$. Note technically it should be $\gamma + \eta$, but $\eta = 0$ in this case with perfect match between concept and neuron. We then test whether a metric passes the test on different values of $\gamma$, using simulated data on tables 4 and 5. Each number is the average result from 1000 evaluations with 500,000 datapoints each. In addition, in Section D, we derived a closed form solution to the binary metrics under missing or extra labels as a function of $\gamma$ and other parameters. This simplifies nicely when we consider an ideal neuron with $a_k = c_t$, and we can derive the expected result of missing/extra labels test as a simple function of $\gamma$ alone in Table 9. These theoretical results perfectly agree with our simulated results.

**Results:** In tables 4 and 5 we ran the simulated results on a variety of different activation sparsities. We can see most metrics (expect from recall and precision) perform well on balanced data ($\gamma = 0.499$ and $\gamma = 0.1$). However, their performance often starts to drop with score difference approaching 0 as the data becomes more and more unbalanced. We can see that practically all the metrics that failed our experimental missing/extra labels tests cannot differentiate between perfect and inperfect concept, specifially on inbalanced data, highlighting that likely the root cause of the failure on these test is that the metrics performs poorly on inbalanced data. This is also aligned with conventional knowledge that metrics such as accuracy and AUC are a poor choice to rely on when your data is heavily inbalanced. On the other hand, metrics that passed the tests are insensitive to activation fequency $\gamma$ and converge to a nonzero constant as $\gamma$ decreases.

We defined a metric as passing the test as having a significant score diff $< 0.01$ on all the evaluated activation frequencies. The results in terms of passing are almost identical to our experimental results in section 4. The only differences were WPMI which passes the theoretical extra labels test but fails the experimental one. We believe this has to do with hyperparameter$(\alpha, \lambda)$ choices and that WPMI can in principle pass the test but with poor hyperparameters it will not, leading us to overall recommend against using it in practice as hyperparamter choice is challenging in the real world. An interesting case is AUPRC and particularly Inverse AUPRC. These metrics are known to be preferable for inbalanced data and work well in that domain, but actually peform worse when the data is balanced, in particular Inverse AUPRC fails the test when data is perfectly balanced. This indicates caution should be used if relying on them, in case you have some neurons with extremely common concepts.

We argue that being able to pass these tests regardless of activation frequency is important for any evaluation metric to be used, as we typically do not know what frequency each neuron will have in advance, in many cases the interesting neurons/concept might activate very sparsely, for example in Sparse autoencoders.

| | Missing Labels Test score diff: | | | | | Pass |
|---|---|---|---|---|---|---|
| **Activation Frequency $\gamma$:** | 0.499 | 0.1 | 0.01 | 0.001 | $\lim_{\gamma \to 0}$ | |
| **Recall** | -0.500 | -0.500 | -0.500 | -0.501 | -0.500 | ✓ |
| **Precision** | 0.000 | 0.000 | 0.000 | 0.000 | 0.000 | × |
| **F1-score** | -0.333 | -0.333 | -0.334 | -0.334 | -0.333 | ✓ |
| **IoU** | -0.500 | -0.500 | -0.500 | -0.501 | -0.500 | ✓ |
| **Accuracy** | -0.250 | -0.050 | -0.005 | -0.001 | 0.000 | × |
| **Balanced Accuracy** | -0.250 | -0.250 | -0.250 | -0.250 | -0.250 | ✓ |
| **Inverse Balanced Accuracy** | -0.166 | -0.026 | -0.003 | 0.000 | 0.000 | × |
| **AUC** | -0.250 | -0.250 | -0.250 | -0.249 | - | ✓ |
| **Inverse AUC** | -0.166 | -0.026 | -0.003 | 0.000 | - | × |
| **Correlation** | -0.211 | -0.156 | -0.147 | -0.147 | - | ✓ |
| **Correlation(top-and-random)** | -0.497 | -0.489 | -0.465 | -0.393 | - | ✓ |
| **Spearman Correlation** | -0.200 | -0.069 | -0.008 | -0.001 | - | × |
| **Spearman Correlation(top-and-random)** | -0.101 | -0.101 | -0.107 | -0.090 | - | ✓ |
| **Cosine** | -0.146 | -0.147 | -0.147 | -0.146 | - | ✓ |
| **Cosine cubed** | -0.240 | -0.147 | -0.147 | -0.147 | - | ✓ |
| **WPMI** | -0.359 | -0.359 | -0.359 | -0.359 | - | ✓ |
| **MAD** | -0.221 | -0.035 | -0.003 | 0.000 | - | × |
| **AUPRC** | -0.251 | -0.450 | -0.495 | -0.500 | - | ✓ |
| **Inverse AUPRC** | -0.500 | -0.500 | -0.500 | -0.499 | - | ✓ |

Table 4: Simulation results missing labels test on idealized neuron with perfect correspondence to a concept activation. We can see most metrics pass when the data is relatively balanced, but start to struggle on inbalanced data (low $\gamma$). The $\lim_{\gamma \to 0}$ column is calculated theoretically.

| | Extra Labels Test score diff: | | | | | Pass |
|---|---|---|---|---|---|---|
| **Activation Frequency $\gamma$:** | 0.499 | 0.1 | 0.01 | 0.001 | $\lim_{\gamma \to 0}$ | |
| **Recall** | 0.000 | 0.000 | 0.000 | 0.000 | 0.000 | ✗ |
| **Precision** | -0.500 | -0.500 | -0.500 | -0.499 | -0.500 | ✓ |
| **F1-score** | -0.333 | -0.333 | -0.333 | -0.334 | -0.333 | ✓ |
| **IoU** | -0.500 | -0.500 | -0.500 | -0.500 | -0.500 | ✓ |
| **Accuracy** | -0.499 | -0.100 | -0.010 | -0.001 | 0.000 | ✗ |
| **Balanced Accuracy** | -0.498 | -0.056 | -0.005 | -0.001 | 0.000 | ✗ |
| **Inverse Balanced Accuracy** | -0.250 | -0.250 | -0.250 | -0.250 | -0.250 | ✓ |
| **AUC** | -0.756 | -0.084 | -0.008 | -0.001 | - | ✗ |
| **Inverse AUC** | -0.250 | -0.250 | -0.250 | -0.250 | - | ✓ |
| **Correlation** | -0.478 | -0.167 | -0.148 | -0.147 | - | ✓ |
| **Correlation(top-and-random)** | -0.493 | -0.029 | -0.003 | 0.000 | - | ✗ |
| **Spearman Correlation** | -0.567 | -0.024 | -0.001 | 0.000 | - | ✗ |
| **Spearman Correlation(top-and-random)** | 0.000 | 0.000 | -0.004 | 0.031 | - | ✗ |
| **Cosine** | -0.146 | -0.146 | -0.146 | -0.146 | - | ✓ |
| **Cosine cubed** | -0.478 | -0.147 | -0.146 | -0.146 | - | ✓ |
| **WPMI** | -0.029 | -0.029 | -0.029 | -0.029 | - | ✓ |
| **MAD** | -0.332 | -0.332 | -0.332 | -0.332 | - | ✓ |
| **AUPRC** | -0.500 | -0.500 | -0.500 | -0.500 | - | ✓ |
| **Inverse AUPRC** | -0.001 | -0.400 | -0.490 | -0.498 | - | ✗ |

Table 5: Simulated extra labels test on idealized neuron with perfect correspondence to a concept activation. We can see most metrics pass when the data is relatively balanced, but start to struggle on inbalanced data (low $\gamma$). The $\lim_{\gamma \to 0}$ column is calculated theoretically.

## C  ADDITIONAL METRIC DEFINITIONS

**F1-score:** F1-score is the harmonic mean of precision and recall, and can be expressed as:

$$M(a_k, c_t) = \frac{2 \cdot B(a_k) \cdot B(c_t)}{||B(a_k)||_1 + ||B(c_t)||_1} \tag{25}$$

**Accuracy:** Standard binary accuracy.

$$M(a_k, c_t) = \frac{TP + TN}{TP + FP + FN + TN} = \frac{B(a_k) \cdot B(c_t) + (1 - B(a_k)) \cdot (1 - B(c_t))}{n} \tag{26}$$

**Balanced Accuracy:** A version of accuracy designed for imbalanced datasets that averages the accuracy on positive and negative inputs.

$$M(a_k, c_t) = \frac{B(a_k) \cdot B(c_t)}{2||B(a_k)||_1} + \frac{(1 - B(a_k)) \cdot (1 - B(c_t))}{2||(1 - B(a_k))||_1} \tag{27}$$

**Inverse Balanced Accuracy:** Balanced accuracy but we consider $a_k$ to be the prediction and $c_t$ to be the ground truth.

$$M(a_k, c_t) = \frac{B(a_k) \cdot B(c_t)}{2||B(c_t)||_1} + \frac{(1 - B(a_k)) \cdot (1 - B(c_t))}{2||(1 - B(c_t))||_1} \tag{28}$$

**Inverse AUC:** Area under receiving-operating-characteristics(ROC) curve, where we consider $a_k$ to be the prediction and $c_t$ to be the ground truth.

$$M(a_k, c_t) = \frac{\sum_{i|B(c_t)_i=0} \sum_{j|B(c_t)_j=1} \mathbb{1}[a_{ki} < a_{kj}] + 0.5 \cdot \mathbb{1}[a_{ki} = a_{kj}]}{||B(c_t)||_1||1 - B(c_t)||_1} \tag{29}$$

**Spearman Correlation:** The Pearson Correlation between the ranks of elements.

$$M(a_k, c_t) = \frac{1}{n} \frac{(R(a_k) - \mu(R(a_k))) \cdot (R(c_t) - \mu(R(c_t)))}{\sigma(R(a_k))\sigma(R(c_t))} \tag{30}$$

**Cosine similarity:** The standard cosine similarity between two vectors.

$$M(a_k, c_t) = \frac{a_k \cdot c_t}{||a_k||_2||c_t||_2} \tag{31}$$

**Cosine Cubed:** A modification of cosine similarity/correlation introduced by [20] that cubes activation hoping to encourage more sensitivity to highest values.

$$M(a_k, c_t) = \frac{[a_k - \mu(a_k)]^3 \cdot [c_t - \mu(c_t)]^3}{||[a_k - \mu(a_k)]^3||_2||[c_t - \mu(c_t)]^3||_2} \tag{32}$$

**WPMI:** Weighted pointwise-mutual information. A version of this objective is used by [3] and [4] to generate explanations, and by [4] to evaluate said explanations.

$$M(a_k, c_t) = \sum_{i|B(a_k)_i=1} [\log(c_{ti}) - \lambda \log(\mu(c_t))] \tag{33}$$

**MAD:** Mean activation difference. Calcualtes the average difference in neuron activations when concept is present vs. when concept is missing.

$$M(a_k, c_t) = \frac{\sum_{i|B(c_t)_i=1} a_{ki}}{||B(c_t)||_1} - \frac{\sum_{j|B(c_t)_j=0} a_{kj}}{||\mathbf{1} - B(c_t)||_1} \tag{34}$$

In the above equations $n$ is the length of $a_k$ and $c_t$, $\mu$ calculates the mean of the vector and $\sigma$ its standard deviation. $\lambda$ is a hyperparameter and $R$ is the rank operator, which transforms each element to its rank, with smallest element becoming 1 and largest $n$.

**AUPRC** Area Under Precision-Recall Curve(AUPRC) is a popular metric for measuring classification performance, in particular for imbalanced data. While we are not aware of a closed form solution, it can be calculated as:

1. Calculate precision and recall at each threshold $\tau_i$, where threshold contains distinct values of $c_t$. Recall $R_i = \frac{B(a_k)\mathbf{1}(c_t \geq \tau_i)}{||B(a_k)||_1}$, precision $P_i = \frac{B(a_k)\mathbf{1}(c_t \geq \tau_i)}{||\mathbf{1}(c_t \geq \tau_i)||_1}$.

2. Calculate area under precision-recall curve using numerical integral:

$$M(a_k, c_t) = \sum_n (R_i - R_{i-1})P_i$$

AUPRC outputs values in $[0, 1]$ range.

**Inverse AUPRC:** Same as AUPRC, but with a differenct framing so we flip $c_t$ and $a_k$ in the calculations.

See Tables 6 and 7 for additional details on our metrics.

| Metric | Definition | Range |
|---|---|---|
| Recall | TP / (TP + FN) | $[0, 1]$ |
| Precision | TP / (TP + FP) | $[0, 1]$ |
| F1-score | 2TP / (2TP + FP + FN) | $[0, 1]$ |
| IoU | TP / (TP + FP + FN) | $[0, 1]$ |
| Accuracy | (TP + TN) / (TP + FP + TN + FN) | $[0, 1]$ |
| Balanced accuracy | [TP / (TP + FN) + TN / (TN + FP)] / 2 | $[0, 1]$ |
| Inverse balanced accuracy (classification version of balanced accuracy (see App. A.2)) | [TP / (TP + FP) + TN / (TN + FN)] | $[0, 1]$ |

Table 6: Definition of commonly-used binary classification metrics. Here, TP, FP, TN, FN refer to true positive, false positive, true negative and false negative, respectively.

| Metric | Definition | Range |
|---|---|---|
| AUC (swap x and y to get inverse AUC) | $$\frac{\sum_{y_i=1}\sum_{y_j=0}[\mathbf{1}\{x_i > x_j\} + 0.5 * \mathbf{1}\{x_i = x_j\}]}{|y=1||y=0|}$$ | $[0,1]$ |
| correlation | $$\frac{\sum(x_i - \bar{x})(y_i - \bar{y})}{\sqrt{\sum(x_i - \bar{x})^2 \sum(y_i - \bar{y})^2}}$$ | $[-1,1]$ |
| Spearman correlation | Replace $x, y$ to corresponding rank $R(x), R(y)$ in correlation | $[-1,1]$ |
| cosine | $$\frac{\sum x_i y_i}{\|x\|_2 \|y\|_2}$$ | $[-1,1]$ |
| cosine cubed | $$\frac{\sum(x_i - \bar{x})^3(y_i - \bar{y})^3}{\|(x - \bar{x})^3\|_2 \|(y - \bar{y})^3\|_2}$$ | $[-1,1]$ |
| WPMI | $\log p(x \mid y) - \lambda \log(p(x))$ | $(-\infty, \infty)$ |
| MAD | $$\frac{\sum_{y_i=1} x_i}{|y_i = 1|} - \frac{\sum_{y_i=0} x_i}{|y_i = 0|}$$ | $(-\infty, \infty)$ |

Table 7: Definition of other commonly-used metrics. Here, $x, y \in \mathbb{R}^N$ are two real vectors, $\bar{x}, \bar{y}$ refer to the mean of $x, y$.

# D    ANALYSIS ON MISSING LABELS/EXTRA LABELS

In this section, we provide a theoratical analysis for missing label and extra label test. For simplicity of symbols, in this section we analyze the population statistics. Suppose we have following confusion matrix:

|       | c=1      | c=0 |
|-------|----------|-----|
| a=1   | $\gamma$ | $b$ |
| a=0   | $\eta$   | $d$ |

In missing labels test, consider a general case where randomly flip $c = 1$ into $c = 0$ with probability $p$. Thus, the resulting confusion matrix is:

|       | c=1              | c=0               |
|-------|------------------|-------------------|
| a=1   | $(1-p)\gamma$    | $b + p\gamma$     |
| a=0   | $(1-p)\eta$      | $d + p\eta$       |

In extra labels test, similarly, we turn $c = 0$ into $c = 1$ with probability $q = \frac{p(\gamma+\eta)}{b+d}$, the resulting confusion matrix is

|       | c=1              | c=0          |
|-------|------------------|--------------|
| a=1   | $\gamma + qb$    | $(1-q)b$     |
| a=0   | $\eta + qd$      | $(1-q)d$     |

With these, we could plug in corresponding TP/FP/TN/FN into metrics to calculate metric value in these two tests.

1. **Recall:**
$$M(a_k, c_t) = \frac{B(a_k) \cdot B(c_t)}{\|B(a_k)\|_1} = \frac{\gamma}{b + \gamma}. \tag{35}$$

   In extra label test:
$$M(a_k, c_t^+) = \frac{\gamma + qb}{b + \gamma} \geq M(a_k, c_t). \tag{36}$$

   In missing label test:
$$M(a_k, c_t^-) = \frac{\gamma - p\gamma}{b + \gamma} \leq M(a_k, c_t). \tag{37}$$

   From the derivation above, we could see that increasing labels only raises recall metric while reducing labels always leads to a drop in recall as we found in our experiments.

2. **Precision:**
$$M(a_k, c_t) = \frac{B(a_k) \cdot B(c_t)}{\|B(c_t)\|_1} = \frac{\gamma}{\gamma + \eta} \tag{38}$$

   In extra label test:
$$M(a_k, c_t^+) = \frac{\gamma + qb}{\gamma + qb + \eta + qd}. \tag{39}$$

   Precision will increase if $\frac{b}{b+d} > \frac{\gamma}{\gamma+\eta}$.

   In missing label test:
$$M(a_k, c_t^-) = \frac{(1-p)\gamma}{(1-p)\gamma + (1-p)\eta} = M(a_k, c_t). \tag{40}$$

   Thus, the precision does not change In missing label test.

3. **F1-score:**
$$M(a_k, c_t) = \frac{2\gamma}{2\gamma + \eta + b} \tag{41}$$

In extra label test:
$$M(a_k, c_t^+) = \frac{2\gamma + 2qb}{2\gamma + qb + \eta + qd + b}. \tag{42}$$

F1-score increase if $\frac{2b}{b+d} > \frac{2\gamma}{2\gamma+\eta+b}$.

In missing label test:
$$M(a_k, c_t^-) = \frac{2(1-p)\gamma}{2(1-p)\gamma + (1-p)\eta + b + p\gamma} = \frac{2\gamma - 2p\gamma}{2\gamma + (1-p)\eta + b - p\gamma}. \tag{43}$$

F1-Score decreases in missing label test.

4. **IoU:**
$$M(a_k, c_t) = \frac{B(a_k) \cdot B(c_t)}{||B(a_k)||_1 + ||B(c_t)||_1 - B(a_k) \cdot B(c_t)} = \frac{\gamma}{\gamma + \eta + b}. \tag{44}$$

In extra label test:
$$M(a_k, c_t^+) = \frac{\gamma + qb}{\gamma + \eta + qd + b}. \tag{45}$$

IoU increases if $\frac{b}{d} > \frac{\gamma}{\gamma+\eta+b}$.

In missing label test:
$$M(a_k, c_t^-) = \frac{(1-p)\gamma}{(1-p)\gamma + (1-p)\eta + b + p\gamma} = \frac{\gamma - p\gamma}{\gamma + (1-p)\eta + b}. \tag{46}$$

Thus, IoU decreases in missing label test.

5. **Accuracy:**
$$M(a_k, c_t) = \frac{B(a_k) \cdot B(c_t) + (\mathbf{1} - B(a_k)) \cdot (\mathbf{1} - B(c_t))}{n} = \gamma + d. \tag{47}$$

In extra label test:
$$M(a_k, c_t^+) = \gamma + qb + d - qd. \tag{48}$$

accuracy increases if $b > d$.

In missing label test:
$$M(a_k, c_t^-) = \gamma - p\gamma + d + p\eta. \tag{49}$$

Accuracy increases if $\eta > \gamma$.

6. **Balanced Accuracy:**
$$M(a_k, c_t) = \frac{B(a_k) \cdot B(c_t)}{2||B(a_k)||_1} + \frac{(\mathbf{1} - B(a_k)) \cdot (\mathbf{1} - B(c_t))}{2||(\mathbf{1} - B(a_k))||_1} = \frac{\gamma}{2\gamma + 2b} + \frac{d}{2\eta + 2d}. \tag{50}$$

In extra label test:
$$M(a_k, c_t^+) = \frac{\gamma + qb}{2\gamma + 2b} + \frac{d - qd}{2\eta + 2d}. \tag{51}$$

balanced accuracy increases if $\frac{b}{2\gamma+2b} > \frac{d}{2\eta+2d}$.

In missing label test:
$$M(a_k, c_t^-) = \frac{\gamma - p\gamma}{2\gamma + 2b} + \frac{d + p\eta}{2\eta + 2d}. \tag{52}$$

balanced accuracy increases if $\frac{\gamma}{2\gamma+2b} < \frac{\eta}{2\eta+2d}$.

7. **Inverse Balanced Accuracy:**
$$M(a_k, c_t) = \frac{B(a_k) \cdot B(c_t)}{2||B(c_t)||_1} + \frac{(\mathbf{1} - B(a_k)) \cdot (\mathbf{1} - B(c_t))}{2||(\mathbf{1} - B(c_t))||_1} = \frac{\gamma}{2\gamma + 2\eta} + \frac{d}{2b + 2d}. \tag{53}$$

In extra label test:
$$M(a_k, c_t^+) = \frac{\gamma + qb}{2\gamma + 2\eta + 2qb + 2qd} + \frac{d - qd}{2b + 2d - 2qd - 2qb}. \tag{54}$$

In missing label test:
$$M(a_k, c_t^-) = \frac{\gamma - p\gamma}{2\gamma + 2\eta - 2p\gamma - 2p\eta} + \frac{d + p\eta}{2b + 2d + 2p\gamma + 2p\eta}. \tag{55}$$

| Metric | Missing label: $M(a_k, c_t^-)$ | Extra label: $M(a_k, c_t^+)$ |
|---|---|---|
| Recall | $1 - p$ | $1$ |
| Precision | $1$ | $\frac{1}{1+p}$ |
| F1-score | $\frac{2-2p}{2-p}$ | $\frac{2}{2+p}$ |
| IoU | $1 - p$ | $\frac{1}{1+p}$ |
| Accuracy | $1 - p\gamma$ | $1 - \gamma p$ |
| Balanced Accuracy | $1 - \frac{p}{2}$ | $1 - \frac{p\gamma}{2(1-\gamma)}$ |
| Inverse Balanced Accuracy | $\frac{(1-\gamma)}{2(1-\gamma)+2p\gamma} + \frac{1}{2}$ | $\frac{2+p}{2(1+p)}$ |

Table 8: The evaluation scores for different metrics under missing and extra label tests on an *ideal* neuron whose activations perfectly match the presence of our concept.

| Metric | Missing labels test($p = 0.5$): $M(a_k, c_t^-) - M(a_k, c_t)$ | Extra labels test($p = 1$): $M(a_k, c_t^+) - M(a_k, c_t)$ |
|---|---|---|
| Recall | $-0.5$ | $0$ |
| Precision | $0$ | $-0.5$ |
| F1-score | $-\frac{1}{3}$ | $-\frac{1}{3}$ |
| IoU | $-0.5$ | $-0.5$ |
| Accuracy | $-\frac{\gamma}{2}$ | $-\gamma$ |
| Balanced Accuracy | $-0.25$ | $-\frac{\gamma}{2(1-\gamma)}$ |
| Inverse Balanced Accuracy | $-\frac{\gamma}{2(2-\gamma)}$ | $-0.25$ |

Table 9: Further simplifying from Table 8 by plugging in $p$ values we typically use in our tests, and $M(a_k, c_t) = 1$, we can calculate the theoretical score diff after running our tests for different binary metrics on ideal neurons.

## D.1 SPECIAL CASE

In this section, we consider a special case where the activation perfectly match the concept, i.e. $c \equiv a$. In this case, we have $\eta = b = 0$, $d = 1 - \gamma$. Plugging in those variables, we can get the following table, which shows how different metrics change after missing-label or extra-label test.

# E  ADDITIONAL RELATED WORKS

**Evaluation of individual neuron explanations.**  Table 10 shows an expanded version of our comparison table (Table 1 in the main text section 4) of existing evaluation methods. It can be seen in Table 10 that prior work [1; 4; 20; 5; 14; 17; 2; 9; 16; 21; 8; 22; 10; 7; 12; 13; 20; 4; 6; 23] only use 1-2 metric for evaluation and did not discuss or justify why the metric should be used. Among all the prior work in Table 10, we believe that the most similar work to ours is [14], which has focused on evaluating individual neuron explanations in language models. In particular, they discover a discrepancy between the evaluation metrics, i.e. neurons with very high correlation(top-and-random) score can still have relatively low F1-scores. However, different from our work their scope is much more specific and they do not provide analysis comparing different evaluation metrics or justification on why they use F1-score specifically. Their finding are in line with ours, where they found that Correlation(top-and-random) fails the extra labels test, while F1-score passes our sanity checks.

Overall we find many evaluations in previous works to be lacking, either due to using poor metrics that fail our sanity checks, or using very small sample sizes. In addition, some popular methods like TCAV [15] completely lack evaluation of whether the concept directions they learn are good. To our best knowledge, no human-study has been conducted using metrics that pass our sanity checks, instead most existing human-studies only measure recall or a similar metric, and running such a study would be valuable for better understanding of unit interpreability and/or explanation methods.

| Metric $M$ | Study | Concept Source $c_t$ | Granularity | Probing Dataset $\mathcal{D}$ | Domain | Target |
|---|---|---|---|---|---|---|
| ~Recall | Human Eval [1] | Crowdsourced | ~Whole Input | Specific eval dataset | Vision | Neurons |
| ~Recall | Human Eval [4; 5] | Crowdsourced | Whole Input | Validation data | Vision | Neurons |
| ~Recall | Human Eval [20] | Crowdsourced | Whole Input | Validation Data | Vision | CBM neurons |
| F1-score | Observation based [14] | Generative + model | Whole Input | Generated + Training data | Language | Neurons |
| F1-score | Sparse probes [17] | Labeled data | Per-token | Specific eval dataset | Language | Linear comb. of neurons |
| IoU | Broden IoU [1; 2; 9] | Labeled data | Per-pixel | Specific eval dataset | Vision | Neurons |
| Accuracy | CBM - concept Error [16] | Labeled data | Whole Input | Validation Data | Vision | CBM neurons |
| ~AUC | Comparative Human Study [21] | Crowdsourced | Whole Input | Training data | Vision | Neurons |
| Inverse AUC | INVERT [8] | Labeled data | Whole Input | Validation data | Vision | Neurons |
| Inverse AUC | CoSy AUC [22] | Generative | Whole Input | Generated+ Validation data | Vision | Neurons |
| Correlation* | Simulation - Correlation Score [10] | Model | Per-token | Training data | Language | Neurons |
| Correlation | Simulation - Correlation Score [7] | Model | Whole Input | Validation data | Vision | Neurons |
| Spearman Correlation* | SAE Auto Interp [12; 13] | Model | Per-token | Training data | Language | SAE features |
| Cosine cubed | LF-CBM - automated [20] | Model | Whole Input | Validation data | Vision | CBM neurons |
| ~WPMI | CLIP-Dissect - Similarity [4] | Model | Whole Input | Validation data | Vision | Neurons |
| MAD | CoSy MAD [22] | Generative | Whole Input | Generated+ Validation data | Vision | Neurons |
| ~MAD | MAIA [6] | Generative | Whole Input | Generated | Vision | Neurons |
| ~MAD | Explanation Score [23] | Generative | Whole Input | Generated | Language | Transformer factors |

Table 10: Extended Table (of Table 1 in the main text section 4) comparing related work.

**Sanity Checks.**  The sanity checks proposed in this paper (Missing and Extra Labels test) are inspired by the sanity checks [24] proposed for saliency maps, which has had a large impact in

guiding that field towards more faithful explanations. However the topic is very different, since [24] focus on local input-importance instead of global neuron explanations in our paper. Besides, the specific tests proposed in [24] are also very different than ours.

**Concept extraction.** Extracting interpretable concepts from a learned representation is a common challenge and relevant for finding individual units to evaluate in our framework. This can either be supervised as proposed by [15] where the concepts are specified by human and labels are provided. This approach is also used by linear probing based work such as [17]. Later, a series of works[25; 26; 27] was proposed to automatically extract concepts from model activations, without human supervision, i.e. unsupervised. [28] claimed that concept extraction could be regarded as a dictionary learning problem. Recently Sparse Autoencoders [19; 13] have also gained popularity as an unsupervised concept extraction method. Note that most unsupervised concept extraction methods are discovering "units" defined in our work that are not directly understandable to humans, and require an explanation method to provide human-understandable explanations. Our work focuses on the evaluation of the explanations of those concepts.

**Concept importance estimation.** Another important task in understanding the behavior of models is estimating the importance of concepts in model decisions. [28] summarized this problem as a general form of attribution methods. However this is typically a local explanation while our framework is focused on global explanations and it is more related to Function 2 defined in Sec. 2.3, i.e. how the unit affects the output.

**Human-centered evaluation on concept-based models.** Human-centered evaluation of model explanation[29; 30] has drawn attention from the XAI community. Recently, [31] collected human evaluation of XAI explanations as a benchmark for explanation methods. These works provide important techniques and inspiration for evaluating explainability, but almost all existing work is focus on local feature importance explanations, which is very different from our work on global neuron-level explanations.

# F ABLATIONS AND EXTRA CHECKS

## F.1 INCREASE/DECREASE FRACTION IN MISSING/EXTRA LABELS TEST

In our standard missing labels we reduce the fraction of positive labels by half, i.e.

$$r_- = \frac{||c_t^-||_1}{||c_t||} = 0.5$$

$$r_+ = \frac{||c_t^+||_1}{||c_t||_1} = 2$$

In this section we run an ablation on the importance of these specific values by running the test on two other combinations of values: $r_- = 0.75$ and $r_+ = 1.33$ (smaller change) and $r_- = 0.33$ and $r_+ = 3$ (larger change). We report the results on final layer neurons (including superclasses) of ViT-B-16 (trained on imagenet) in Tables 11 and 12. Overall we can see that these parameters do not impact our qualitative observations, i.e. which metric passes vs which doesn't. While Avg score diff changes proportionally to $r$, it doesn't affect which metrics are close to zero vs which are not, and decrease acc remains relatively unchanged. Overall this shows our sanity test is not sensitive to the specific parameter choice but instead reflects overall trends of the metric.

| | ViT-B-16 final layer neurons, including superclasses | | | |
|---|---|---|---|---|
| | **Missing Labels Test:** $r_- = 0.75$ | | **Extra Labels Test:** $r_+ = 1.333$ | |
| **Metric** | **Avg Score Diff** | **Decrease acc** | **Avg Score Diff** | **Decreased acc** |
| **Recall** | -0.2367 | 73.83% | 0.0000 | 0.00% |
| **Precision** | -0.0009 | 51.73% | -0.1628 | 99.92% |
| **F1-score** | -0.0946 | 94.14% | -0.1040 | 99.92% |
| **IoU** | -0.1241 | 94.14% | -0.1310 | 99.92% |
| **Accuracy** | 0.0008 | 97.59% | -0.0018 | 99.92% |
| **Balanced Accuracy** | -0.1174 | 73.68% | -0.0009 | 99.92% |
| **Inverse Balanced Acc.** | 0.0002 | 49.92% | -0.0198 | 99.92% |
| **AUC** | -0.1160 | 79.25% | -0.0008 | 75.71% |
| **Inverse AUC** | -0.0014 | 87.37% | -0.1248 | 99.92% |
| **Correlation** | -0.0591 | 99.70% | -0.0582 | 99.92% |
| **Correlation (top-and-random)** | -0.0798 | 90.45% | -0.0001 | 48.42% |
| **Spearman Correlation** | -0.0017 | 59.92% | -0.0006 | 49.25% |
| **Spearman Correlation (top-and-random)** | -0.0658 | 80.60% | -0.0020 | 51.20% |
| **Cosine** | -0.0582 | 99.77% | -0.0565 | 99.92% |
| **Cosine cubed** | -0.0565 | 98.87% | -0.0559 | 99.92% |
| **WPMI** | -0.1604 | 93.23% | -0.0110 | 99.92% |
| **MAD** | -0.0020 | 55.56% | -0.0911 | 99.70% |
| **AUPRC** | -0.1418 | 99.17% | -0.1386 | 99.92% |
| **Inverse AUPRC** | -0.2044 | 99.77% | -0.2088 | 99.70% |

Table 11: Sanity check results with smaller change in labels.

| ViT-B-16 final layer neurons, including superclasses | | | | |
|---|---|---|---|---|
| **Metric** | **Missing Labels Test:** $r_- = 0.33$ | | **Extra Labels Test:** $r_+ = 3$ | |
| | **Avg Score Diff** | **Decrease acc** | **Avg Score Diff** | **Decreased acc** |
| **Recall** | -0.6297 | 99.40% | 0.0000 | 0.00% |
| **Precision** | 0.0002 | 47.44% | -0.4404 | 99.92% |
| **F1-score** | -0.3297 | 97.14% | -0.3580 | 99.92% |
| **IoU** | -0.3516 | 97.14% | -0.3718 | 99.92% |
| **Accuracy** | 0.002 | 97.67% | -0.0098 | 99.92% |
| **Balanced Accuracy** | -0.3122 | 99.10% | -0.0050 | 99.92% |
| **Inverse Balanced Acc.** | 0.001 | 52.86% | -0.0535 | 99.92% |
| **AUC** | -0.3102 | 98.50% | -0.0050 | 90.98% |
| **Inverse AUC** | -0.0022 | 78.87% | -0.3363 | 99.92% |
| **Correlation** | -0.1824 | 99.92% | -0.1842 | 99.92% |
| **Correlation (top-and-random)** | -0.2166 | 99.40% | -0.0049 | 52.11% |
| **Spearman Correlation** | -0.0045 | 67.97% | -0.0023 | 50.60% |
| **Spearman Correlation (top-and-random)** | -0.1793 | 97.14% | -0.0058 | 53.08% |
| **Cosine** | -0.1817 | 100.00% | -0.1802 | 99.92% |
| **Cosine cubed** | -0.1753 | 99.92% | -0.1766 | 99.92% |
| **WPMI** | -0.4169 | 100.00% | -0.0430 | 99.77% |
| **MAD** | -0.0034 | 53.23% | -0.2450 | 99.70% |
| **AUPRC** | -0.3715 | 99.85% | -0.3750 | 99.92% |
| **Inverse AUPRC** | -0.5387 | 100.00% | -0.5599 | 99.32% |

Table 12: Sanity check results with larger change in labels.

## F.2 FAILURE CASE OF COSINE: ADDING A CONSTANT ACTIVATION

In this section we dicuss a failure case of cosine similarity as an evaluation metric. The main idea is that cosine similarity outputs are not independent of the mean of the neuron's activation, while all other evaluation metrics are. This causes it to associate neurons with large average activation values with very generic concepts that are almost always active. As an example, we added a constant (1) to all activations of concept neurons in a Concept Bottleneck Model [16] trained on CUB200, and ran our argmax generation test. All other evaluation metrics are invariant to this change, but the accuracy of Cosine drops from 99.07% to 0.93% with this small change, as shown in Table 13. We believe this is a flaw that points against using cosine similarity, as the average activation of a hidden layer neuron is not functionally important, and could be absorbed into biases of the next layer. Instead we recommend using Pearson correlation which is identical to cosine similarity after normalizing to mean 0, as we show in Section A.3.

| | CUB200 - CBM concept neurons | | |
| | Original $a_k$ | $a_k' = a_k + 1$ | |
| **Metric** | **Argmax acc** | **Argmax acc** | $\Delta$ **acc** |
| **Recall** | 49.25% | 49.25% | 0.00% |
| **Precision** | 84.11% | 84.11% | 0.00% |
| **F1-score/IoU** | 77.57% | 77.57% | 0.00% |
| **Accuracy** | 81.78% | 81.78% | 0.00% |
| **Balanced Accuracy** | 77.57% | 77.57% | 0.00% |
| **Inverse Balanced Accuracy** | 69.16% | 69.16% | 0.00% |
| **AUC** | 81.31% | 81.31% | 0.00% |
| **Inverse AUC** | **99.07%** | **99.07%** | 0.00% |
| **Correlation(full)** | **99.07%** | **99.07%** | 0.00% |
| **Correlation(top-and-random)** | 86.56% | 86.56% | 0.00% |
| **Spearman Correlation(full)** | 54.21% | 54.21% | 0.00% |
| **Spearman Correlation(top-and-random)** | 62.62% | 62.62% | 0.00% |
| **Cosine** | **99.07%** | 0.93% | -98.14% |
| **Cosine cubed** | **99.07%** | **99.07%** | 0.00% |
| **WPMI** | 80.37% | 80.37% | 0.00% |
| **MAD** | **99.07%** | **99.07%** | 0.00% |
| **AUPRC** | 71.03% | 71.03% | 0.00% |
| **Inverse AUPRC** | 31.78% | 31.78% | 0.00% |

Table 13: Adding a constant to the activation values of all neurons causes the cosine similarity to perform very poorly, while other metrics are unchanged.

# G DETAILED RESULTS

## G.1 EXPERIMENTAL SETUP DETAILS

**Missing and Extra labels test:**   We evaluate our experimental results across 6 settings:

1. Dataset $\mathcal{D}$=ImageNet, ViT-B-16 [32] trained on imagenet, final layer neurons(and super-class neurons), Table 14

2. Dataset $\mathcal{D}$=ImageNet, ResNet-50 trained on imagenet, layer4 neurons, Table 15

3. Dataset $\mathcal{D}$=Places365, ResNet-18 trained on Places365, final layer neurons, Table 16

4. Dataset $\mathcal{D}$=Places365, ResNet-18 trained on Places365, layer4 neurons, Table 17

5. Dataset $\mathcal{D}$=CUB200, CBM trained on CUB200, concept neurons, Table 18

6. Dataset $\mathcal{D}$=CUB200, CLIP ViT-B-32 image encoder, linear probe trained to detect CUB-concepts, Table 19

For all settings we used the ground truth labels from the dataset as $c_t$. The ImageNet [33] and Places [34] models were pretrained. For CUB-CBM we trained our own model using the code released by [16]. Our CBM reached 96.75% concept accuracy on the test set which is in line with their reported results. For CLIP, we used the pretrained model from [35], and then learned a linear probe on top of frozen image embeddings to minimize binary cross-entropy loss on the training split of CUB200[36], with early stopping using validation data. Our linear probe reached 89.76% concept accuracy. The CUB dataset is a small bird species classification dataset that contains detailed annotations for lower level concepts, such as wing color. Following [16], we only used the 112 concepts that are present on at least 5% of the inputs and our CLIP linear probe was trained to predict these concepts, not the final class of inputs.

For the final layer neurons as well as CUB neurons we let "correct" concept $t_k$ be the ground truth concept for that neuron. We choose the hyperparameter $\alpha$ that maximizes AUC(Eq. 19) performance on validation neurons, and run the tests on test neurons. For all evaluations we used neuron activations after the activation function (i.e. softmax/sigmoid).

While for layer4(after avg pool) neurons we defined the "correct" $t_k$ correct" concept $t_k$ as the concept that maximizes IoU with $\alpha = 0.005$ similar to [1]. For these layers we fixed $\alpha = 0.005$ for all metrics as that was used to determine the "ground truth".

Interestingly, we find that most methods pass the tests on the CUB dataset than the other datasets we looked at, but the trends in terms of which metrics perform worse are still similar. We believe this is caused by data imbalance, as the concepts in CUB are relatively balanced (following [16] we only keep concepts that are present on at least 5% of the inputs), while for example ImageNet classes are much more imbalanced (each class is positive on 0.1% of the inputs). This is confirmed by our theoretical observations in Section B, which show that poor metrics are much more likely to fail the test when the concepts are imbalanced.

**Argmax generation and AUC**   We evaluate the argmax generation and AUC test on 8 different settings:

1. Dataset $\mathcal{D}$=ImageNet, ViT-B-16(ImageNet), final layer neurons, ground truth $c_t$

2. Dataset $\mathcal{D}$=ImageNet, ViT-B-16(ImageNet), final layer neurons, SigLIP $c_t$

3. Dataset $\mathcal{D}$=ImageNet, ViT-B-16(ImageNet), final layer neurons+superclass neurons, ground truth $c_t$

4. Dataset $\mathcal{D}$=ImageNet, ViT-B-16(ImageNet), final layer neurons+superclass neurons, SigLIP $c_t$

5. Dataset $\mathcal{D}$=Places365, ResNet-18(Places365), final layer neurons, ground truth $c_t$

6. Dataset $\mathcal{D}$=Places365, ResNet-18(Places365), final layer neurons, SigLIP $c_t$

7. Dataset $\mathcal{D}$=CUB200, CBM trained on CUB200, concept neurons, ground truth $c_t$

8. Dataset $\mathcal{D}$=CUB200, CLIP ViT-B-32 image encoder, linear probe trained to detect CUB-concepts, ground truth $c_t$

SigLIP $c_t$ indicates we used Pseudo-labels generated from SigLIP (ViT-SO400M-14-SigLIP-384) [37] as done by [7]. For all metrics evaluations we choose hyperparameters such as $\alpha$ by finding the one with best performance on validation neurons (random subset of 5% of the neurons), and use those hyperparameters to evaluate on test neurons.

Tables 20 and 21 show the detailed results of our argmax generation experiment, and Tables 22 and 23 show the detailed results of our AUC evaluation experiment. We can see that overall the AUC numbers are quite high for most methods, as the dataset is very imbalanced with more examples of incorrect pairs than correct pairs, and the task of telling a correct explanation apart from a random one is quite easy. However we can still find meaningful differences between metrics by precisely measuring how close they can get to perfect score of 1.

| | ViT-B-16 final layer neurons, including superclass | | | |
| | Missing Labels Test | | Extra Labels Test | |
| | Avg Score Diff | Decrease acc | Avg Score Diff | Decreased acc |
|---|---|---|---|---|
| **Recall** | -0.4772 | 96.02% | 0.0000 | 0.00% |
| **Precision** | 0.0019 | 46.92% | -0.3287 | 99.92% |
| **F1-score** | -0.2191 | 96.77% | -0.2404 | 99.92% |
| **IoU** | -0.2557 | 96.77% | -0.2719 | 99.92% |
| **Accuracy** | 0.0016 | 97.59% | -0.0053 | 99.92% |
| **Balanced Accuracy** | -0.2366 | 95.79% | -0.0027 | 99.92% |
| **Inverse Balanced Acc.** | 0.0014 | 50.08% | -0.0392 | 99.92% |
| **AUC** | -0.2378 | 95.11% | -0.0026 | 88.05% |
| **Inverse AUC** | -0.0021 | 80.83% | -0.2495 | 99.92% |
| **Correlation** | -0.1268 | 99.92% | -0.1272 | 99.92% |
| **Correlation (top-and-random)** | -0.1563 | 98.12% | -0.0023 | 50.23% |
| **Spearman Correlation** | -0.0034 | 64.59% | -0.0014 | 50.60% |
| **Spearman Correlation (top-and-random)** | -0.1355 | 93.61% | -0.0039 | 51.88% |
| **Cosine** | -0.1260 | 100.00% | -0.1245 | 99.92% |
| **Cosine cubed** | -0.1217 | 99.77% | -0.1222 | 99.92% |
| **WPMI** | -0.3158 | 99.85% | -0.0271 | 99.92% |
| **MAD** | -0.0016 | 51.28% | -0.1839 | 99.70% |
| **AUPRC** | -0.2791 | 99.85% | -0.2797 | 99.92% |
| **Inverse AUPRC** | -0.4041 | 100.00% | -0.4200 | 99.62% |

Table 14: Missing and Extra Labels test results on ViT-B-16(ImageNet) final layer neurons.

| Resnet-50 layer4 neurons | | | | |
|---|---|---|---|---|
| | Missing Labels Test | | Extra Labels Test | |
| | Avg Score Diff | Decrease acc | Avg Score Diff | Decreased acc |
| Recall | -0.0814 | 100.00% | 0.0021 | 0.00% |
| Precision | 0.0027 | 47.94% | -0.2122 | 100.00% |
| F1-score | -0.0834 | 100.00% | -0.0434 | 100.00% |
| IoU | -0.0506 | 100.00% | -0.0275 | 100.00% |
| Accuracy | 0.0006 | 34.07% | -0.0027 | 100.00% |
| Balanced Accuracy | -0.0402 | 100.00% | -0.0003 | 71.12% |
| Inverse Balanced Acc. | 0.0012 | 49.74% | -0.1061 | 100.00% |
| AUC | -0.0401 | 88.23% | -0.0004 | 50.72% |
| Inverse AUC | -0.0002 | 52.67% | -0.2214 | 100.00% |
| Correlation | -0.0270 | 100.00% | -0.0264 | 100.00% |
| Correlation (top-and-random) | -0.0648 | 85.61% | -0.0030 | 51.80% |
| Spearman Correlation | 0.0067 | 69.22% | 0.0000 | 50.77% |
| Spearman Correlation (top-and-random) | -0.0443 | 67.83% | 0.0012 | 49.28% |
| Cosine | -0.0237 | 100.00% | -0.0114 | 98.51% |
| Cosine cubed | -0.0427 | 99.33% | -0.0406 | 100.00% |
| WPMI | -0.0384 | 88.03% | -0.0320 | 100.00% |
| MAD | 0.0003 | 50.77% | -0.1163 | 100.00% |
| AUPRC | -0.0325 | 99.49% | -0.0325 | 100.00% |
| Inverse AUPRC | -0.0809 | 98.30% | -0.0809 | 98.30% |

Table 15: Missing and Extra labels test results on ResNet-50(Imagenet) layer4 neurons.

| Resnet-18(Places365) - Final layer neurons | | | | |
|---|---|---|---|---|
| | Missing Labels Test | | Extra Labels Test | |
| | Avg Score Diff | Decrease acc | Avg Score Diff | Decreased acc |
| Recall | -0.4301 | 97.98% | 0 | 0.00% |
| Precision | -0.0039 | 53.03% | -0.0324 | 98.85% |
| F1-score | -0.0151 | 63.40% | -0.0577 | 98.85% |
| IoU | -0.0079 | 63.40% | -0.0319 | 98.85% |
| Accuracy | 0.0012 | 0.00% | -0.0027 | 100.00% |
| Balanced Accuracy | -0.2144 | 97.98% | -0.0014 | 100.00% |
| Inverse Balanced Acc. | -0.0020 | 55.33% | -0.0162 | 100.00% |
| AUC | -0.1976 | 98.56% | -0.0021 | 61.96% |
| Inverse AUC | -0.0008 | 66.57% | -0.2435 | 100.00% |
| Correlation | -0.0894 | 99.71% | -0.0883 | 100.00% |
| Correlation (top-and-random) | -0.1249 | 95.39% | -0.0026 | 50.43% |
| Spearman Correlation | -0.0018 | 69.45% | 0.0002 | 48.99% |
| Spearman Correlation (top-and-random) | -0.1267 | 90.78% | 0.0004 | 51.30% |
| Cosine | -0.0895 | 99.71% | -0.0873 | 100.00% |
| Cosine cubed | -0.0780 | 99.14% | -0.0755 | 99.71% |
| WPMI | -0.3084 | 99.42% | 0.0013 | 0.00% |
| MAD | -0.0009 | 48.99% | -0.2264 | 100.00% |
| AUPRC | -0.0270 | 85.30% | -0.0282 | 98.85% |
| Inverse AUPRC | -0.2578 | 100.00% | -0.2621 | 99.71% |

Table 16: Missing and Extra labels test results on ResNet-18(Places365) final layer neurons.

| | Resnet-18(Places365) - layer4 neurons | | | |
| --- | --- | --- | --- | --- |
| | Missing Labels Test | | Extra Labels Test | |
| | Avg Score Diff | Decrease acc | Avg Score Diff | Decreased acc |
| Recall | -0.0843 | 100.00% | 0.0022 | 0.00% |
| Precision | 0.0028 | 45.79% | -0.1537 | 100.00% |
| F1-score | -0.0855 | 99.79% | -0.0552 | 100.00% |
| IoU | -0.0540 | 99.79% | -0.0358 | 100.00% |
| Accuracy | 0.0005 | 10.88% | -0.0027 | 100.00% |
| Balanced Accuracy | -0.0417 | 100.00% | -0.0003 | 66.12% |
| Inverse Balanced Acc. | 0.0012 | 46.00% | -0.0768 | 100.00% |
| AUC | -0.0437 | 88.50% | -0.0013 | 52.98% |
| Inverse AUC | -0.0008 | 53.80% | -0.2123 | 100.00% |
| Correlation | -0.0266 | 100.00% | -0.0267 | 100.00% |
| Correlation (top-and-random) | -0.0623 | 83.78% | -0.0058 | 52.16% |
| Spearman Correlation | -0.0020 | 72.28% | -0.0005 | 54.21% |
| Spearman Correlation (top-and-random) | -0.0520 | 70.84% | 0.0034 | 50.92% |
| Cosine | -0.0251 | 100.00% | -0.0131 | 100.00% |
| Cosine cubed | -0.0395 | 99.79% | -0.0377 | 100.00% |
| WPMI | -0.0940 | 91.58% | -0.0713 | 100.00% |
| MAD | 0.0005 | 47.84% | -0.1612 | 100.00% |
| AUPRC | -0.0306 | 98.36% | -0.0315 | 100.00% |
| Inverse AUPRC | -0.0698 | 99.59% | -0.0700 | 100.00% |

Table 17: Missing and Extra labels test results on ResNet-18(Places365) layer4 neurons.

| | CBM(CUB200) - concept neurons | | | |
| --- | --- | --- | --- | --- |
| | Missing Labels Test | | Extra Labels Test | |
| | Avg Score Diff | Decrease acc | Avg Score Diff | Decreased acc |
| Recall | -0.5002 | 100.00% | 0.0012 | 0.00% |
| Precision | -0.0003 | 48.60% | -0.2993 | 100.00% |
| F1-score | -0.1990 | 100.00% | -0.2290 | 100.00% |
| IoU | -0.2150 | 100.00% | -0.2365 | 100.00% |
| Accuracy | -0.0884 | 100.00% | -0.0732 | 98.13% |
| Balanced Accuracy | -0.2054 | 100.00% | -0.0943 | 100.00% |
| Inverse Balanced Acc. | -0.0424 | 100.00% | -0.1770 | 100.00% |
| AUC | -0.2042 | 100.00% | -0.0926 | 100.00% |
| Inverse AUC | -0.0610 | 100.00% | -0.2850 | 100.00% |
| Correlation | -0.1541 | 100.00% | -0.1918 | 100.00% |
| Correlation (top-and-random) | -0.1936 | 99.07% | -0.1245 | 93.46% |
| Spearman Correlation | -0.0977 | 100.00% | -0.0720 | 91.59% |
| Spearman Correlation (top-and-random) | -0.1534 | 93.46% | -0.1229 | 90.65% |
| Cosine | -0.1349 | 100.00% | -0.1227 | 100.00% |
| Cosine cubed | -0.1467 | 100.00% | -0.1950 | 100.00% |
| WPMI | -0.4273 | 100.00% | -0.0228 | 100.00% |
| MAD | -0.0593 | 95.33% | -0.2300 | 99.07% |
| AUPRC | -0.2172 | 100.00% | -0.2527 | 100.00% |
| Inverse AUPRC | -0.4749 | 100.00% | -0.2212 | 93.46% |

Table 18: Missing and Extra labels test results on CBM(CUB200) concept neurons.

| | CLIP ViT-B-16 - Linear probe for CUB200 concepts | | | |
|---|---|---|---|---|
| | Missing Labels Test | | Extra Labels Test | |
| | Avg Score Diff | Decrease acc | Avg Score Diff | Decreased acc |
| **Recall** | -0.4534 | 100.00% | 0.0138 | 0.00% |
| **Precision** | -0.0008 | 52.34% | -0.2698 | 100.00% |
| **F1-score** | -0.1615 | 100.00% | -0.1739 | 100.00% |
| **IoU** | -0.1462 | 100.00% | -0.1538 | 100.00% |
| **Accuracy** | -0.0762 | 100.00% | -0.0588 | 96.26% |
| **Balanced Accuracy** | -0.1794 | 100.00% | -0.0855 | 100.00% |
| **Inverse Balanced Acc.** | -0.0361 | 100.00% | -0.1610 | 100.00% |
| **AUC** | -0.1651 | 100.00% | -0.0831 | 98.13% |
| **Inverse AUC** | -0.0519 | 100.00% | -0.2524 | 100.00% |
| **Correlation** | -0.1172 | 100.00% | -0.1457 | 100.00% |
| **Correlation (top-and-random)** | -0.1622 | 98.13% | -0.1050 | 94.39% |
| **Spearman Correlation** | -0.0835 | 100.00% | -0.0593 | 88.79% |
| **Spearman Correlation (top-and-random)** | -0.1379 | 95.33% | -0.1086 | 85.05% |
| **Cosine** | -0.1111 | 100.00% | -0.0634 | 97.20% |
| **Cosine cubed** | -0.1061 | 100.00% | -0.1388 | 100.00% |
| **WPMI** | -0.3796 | 100.00% | -0.0202 | 90.65% |
| **MAD** | -0.0379 | 97.20% | -0.1354 | 96.26% |
| **AUPRC** | -0.1394 | 100.00% | -0.1592 | 100.00% |
| **Inverse AUPRC** | -0.3711 | 100.00% | -0.1020 | 74.77% |

Table 19: Missing and Extra labels test results on linear probe for CUB concepts trained on CLIP embeddings.

| | Setup 1: gt $c_t$, | | Setup 2: SigLIP $c_t$ | | Setup 3: gt $c_t$, | | Setup 4: SigLIP $c_t$ | |
|---|---|---|---|---|---|---|---|---|
| | original $K$ and $C$ | | original $K$ and $C$ | | original+superclass $K$ and $C$ | | original+superclass $K$ and $C$ | |
| Metric | Accuracy | Rank | Accuracy | Rank | Accuracy | Rank | Accuracy | Rank |
| **Recall** | 99.37% | 6 | 94.42% | 10 | 8.33% | 16 | 67.43% | 11 |
| **Precision** | 99.37% | 6 | 93.90% | 13 | 71.81% | 13 | 65.75% | 13 |
| **F1-score/IoU** | 99.37% | 6 | 94.32% | 11 | 88.90% | 5 | 68.46% | 10 |
| **Accuracy** | 99.37% | 6 | 92.24% | 14 | 82.59% | 9 | 65.43% | 15 |
| **Balanced Accuracy** | 99.37% | 6 | 94.51% | 9 | 83.53% | 8 | 68.69% | 8 |
| **Inverse Balanced Acc.** | 99.37% | 6 | 93.98% | 12 | 72.97% | 11 | 65.75% | 13 |
| **AUC** | 98.84% | 14 | 96.79% | 5 | 78.57% | 10 | 75.30% | 4 |
| **Inverse AUC** | 94.74% | 16 | 85.68% | 16 | 85.41% | 6 | 57.56% | 16 |
| **Correlation** | **99.58%** | **1** | **98.11%** | **1** | 99.25% | 2 | 76.95% | 2 |
| **Correlation (top-and-random)** | 98.74% | 15 | 87.79% | 15 | 68.94% | 14 | 68.50% | 9 |
| **Spearman Correlation** | 0.74% | 18 | 10.63% | 18 | 2.48% | 17 | 7.37% | 18 |
| **Spearman Correlation (top-and-random)** | 64.95% | 17 | 71.84% | 17 | 23.46% | 15 | 49.44% | 17 |
| **Cosine** | **99.58%** | **1** | **98.11%** | **1** | **99.32%** | **1** | 76.84% | 3 |
| **Cosine cubed** | 99.47% | 5 | **98.11%** | **1** | 98.65% | 3 | 74.77% | 5 |
| **WPMI** | 99.37% | 6 | 96.74% | 6 | 85.41% | 6 | **77.11%** | **1** |
| **MAD** | **99.58%** | **1** | 96.11% | 7 | 72.41% | 12 | 66.95% | 12 |
| **AUPRC** | 99.37% | 6 | 96.95% | 4 | 94.40% | 4 | 72.29% | 6 |
| **Inverse AUPRC** | **99.58%** | **1** | 94.63% | 8 | 0.45% | 18 | 69.62% | 7 |

Table 20: Detailed results of our argmax evaluation experiment on final layer neurons of ViT-B-16(ImageNet).

| Metric | Setup 5: gt $c_t$, Resnet-18 (Places365) Accuracy | Rank | Setup 6: SigLIP $c_t$, Resnet-18 (Places365) Accuracy | Rank | Setup 7: gt $c_t$, CBM (CUB200) Accuracy | Rank | Setup 8: gt $c_t$, CLIP linear probe (CUB200) Accuracy | Rank |
|---|---|---|---|---|---|---|---|---|
| Recall | 97.12% | 6 | 84.08% | 13 | 49.25% | 17 | 11.33% | 17 |
| Precision | 97.12% | 6 | 85.31% | 11 | 84.11% | 7 | 49.22% | 14 |
| F1-score/IoU | 97.12% | 6 | 85.46% | 9 | 77.57% | 11 | 64.49% | 7 |
| Accuracy | 97.12% | 6 | 77.54% | 14 | 81.78% | 8 | 57.94% | 10 |
| Balanced Accuracy | 97.12% | 6 | 85.43% | 10 | 77.57% | 11 | 62.62% | 8 |
| Inverse Balanced Acc. | 97.12% | 6 | 85.47% | 8 | 69.16% | 14 | 69.16% | 5 |
| AUC | 94.52% | 16 | 85.59% | 7 | 81.31% | 9 | 52.34% | 12 |
| Inverse AUC | 94.81% | 15 | 76.95% | 15 | **99.07%** | **1** | 84.11% | 3 |
| Correlation | **98.27%** | **1** | 92.51% | 2 | **99.07%** | **1** | **86.92%** | **1** |
| Correlation (top-and-random) | 97.12% | 6 | 70.03% | 16 | 86.56% | 6 | 54.55% | 11 |
| Spearman Correlation | 4.32% | 18 | 34.29% | 18 | 54.21% | 16 | 34.58% | 15 |
| Spearman Correlation (top-and-random) | 74.06% | 17 | 61.96% | 17 | 62.62% | 15 | 24.30% | 16 |
| Cosine | **98.27%** | **1** | 92.51% | 2 | **99.07%** | **1** | 81.31% | 4 |
| Cosine cubed | 97.98% | 5 | **93.37%** | **1** | **99.07%** | **1** | 85.98% | 2 |
| WPMI | 97.12% | 6 | 85.01% | 12 | 80.37% | 10 | 50.47% | 13 |
| MAD | **98.27%** | **1** | 89.63% | 5 | **99.07%** | **1** | 67.29% | 6 |
| AUPRC | 97.12% | 6 | 91.93% | 4 | 71.03% | 13 | 58.41% | 9 |
| Inverse AUPRC | **98.27%** | **1** | 86.46% | 6 | 31.78% | 18 | 2.80% | 18 |

Table 21: Detailed results of our argmax evaluation experiment on final layer neurons of Places365 models and concept neurons on CUB200.

| Metric | Setup 1: gt $c_t$ original $K$ and $C$ AUC | Rank | Setup 2: SigLIP $c_t$ original $K$ and $C$ AUC | Rank | Setup 3: gt $c_t$ original+superclass $K$ and $C$ AUC | Rank | Setup 4: SigLIP $c_t$ original+superclass $K$ and $C$ AUC | Rank |
|---|---|---|---|---|---|---|---|---|
| Recall | 0.9999993 | 6 | 0.9877944 | 15 | 0.9960657 | 14 | 0.9432574 | 16 |
| Precision | 0.9999993 | 6 | 0.9820718 | 17 | 0.9996042 | 5 | 0.9553497 | 12 |
| F1-score/IoU | 0.9999993 | 6 | 0.9820726 | 16 | 0.9996820 | 4 | 0.9435605 | 15 |
| Accuracy | 0.9999993 | 6 | 0.9991594 | 7 | 0.9765614 | 17 | 0.9893615 | 6 |
| Balanced Accuracy | 0.9999993 | 6 | 0.9976398 | 9 | 0.9994295 | 8 | 0.9858313 | 9 |
| Inverse Balanced Acc. | 0.9999993 | 6 | 0.9968502 | 11 | 0.9989645 | 10 | 0.9854770 | 10 |
| AUC | 0.9999915 | 15 | 0.9997286 | 6 | 0.9959190 | 15 | 0.9868359 | 7 |
| Inverse AUC | 0.9999157 | 16 | 0.9902301 | 13 | 0.9990078 | 9 | 0.9510220 | 13 |
| Correlation | **0.9999998** | **1** | 0.9999713 | 3 | 0.9999300 | 2 | 0.9905589 | 4 |
| Correlation (top-and-random) | 0.9999993 | 6 | 0.9981871 | 8 | 0.9982840 | 11 | 0.9859906 | 8 |
| Spearman Correlation | 0.6809415 | 18 | 0.7629487 | 18 | 0.7471902 | 18 | 0.7794497 | 18 |
| Spearman Correlation (top-and-random) | 0.9928371 | 17 | 0.9974837 | 10 | 0.9917333 | 16 | 0.9839782 | 11 |
| Cosine | **0.9999998** | **1** | 0.9999708 | 4 | **0.9999481** | **1** | **0.9974763** | **1** |
| Cosine cubed | **0.9999998** | **1** | 0.9999729 | 2 | 0.9995726 | 6 | 0.9915143 | 3 |
| WPMI | 0.9999954 | 14 | 0.9998929 | 5 | 0.9995661 | 7 | 0.9918070 | 2 |
| MAD | 0.9999994 | 5 | 0.9941177 | 12 | 0.9982341 | 12 | 0.9453136 | 14 |
| AUPRC | 0.9999993 | 6 | **0.9999745** | **1** | 0.9997836 | 3 | 0.9902171 | 5 |
| Inverse AUPRC | **0.9999998** | **1** | 0.9896386 | 14 | 0.9975076 | 13 | 0.9352833 | 17 |

Table 22: Detailed results of our AUC evaluation on Final layer neurons of ViT-B-16(ImageNet).

| Method | Setup 5: gt $c_t$, Resnet-18 (Places365) AUC | Rank | Setup 6: SigLIP $c_t$ Resnet-18 (Places365) AUC | Rank | Setup 7: gt $c_t$, CBM (CUB200) AUC | Rank | Setup 8: gt $c_t$, CLIP linear probe (CUB200) AUC | Rank |
|---|---|---|---|---|---|---|---|---|
| **Recall** | 0.9941722 | 12 | 0.9716787 | 17 | 0.9932254 | 10 | 0.9603242 | 14 |
| **Precision** | 0.9941722 | 12 | 0.9759407 | 13 | 0.9901683 | 12 | 0.9628336 | 13 |
| **F1-score/IoU** | 0.9941722 | 12 | 0.9719225 | 16 | 0.9832933 | 15 | 0.9711482 | 12 |
| **Accuracy** | 0.9942055 | 9 | 0.9870149 | 11 | 0.9574415 | 18 | 0.9380949 | 18 |
| **Balanced Accuracy** | 0.9942055 | 9 | 0.9882966 | 10 | 0.9906365 | 11 | 0.9768571 | 7 |
| **Inverse Balanced Acc.** | 0.9942055 | 9 | 0.9913224 | 9 | 0.9965389 | 8 | 0.9859675 | 2 |
| **AUC** | 0.9930562 | 16 | 0.9949677 | 6 | 0.9898162 | 13 | 0.9753796 | 9 |
| **Inverse AUC** | 0.9993843 | 2 | 0.9742231 | 15 | **0.9995459** | **1** | 0.9853412 | 4 |
| **Correlation** | 0.9993321 | 3 | 0.9989897 | 2 | 0.9995003 | 2 | **0.9905228** | **1** |
| **Correlation (top-and-random)** | 0.9944704 | 8 | 0.9929705 | 8 | 0.9972774 | 7 | 0.9824368 | 6 |
| **Spearman Correlation** | 0.8586794 | 18 | 0.9522357 | 18 | 0.9711632 | 17 | 0.9506609 | 17 |
| **Spearman Correlation (top-and-random)** | 0.9787217 | 17 | 0.9937719 | 7 | 0.9777333 | 16 | 0.9541921 | 16 |
| **Cosine** | 0.9993151 | 4 | 0.9988518 | 3 | 0.9993839 | 4 | 0.9848683 | 5 |
| **Cosine cubed** | 0.9989203 | 6 | 0.9986860 | 4 | 0.9994051 | 3 | 0.9857802 | 3 |
| **WPMI** | 0.9972885 | 7 | 0.9972979 | 5 | 0.9983350 | 6 | 0.9745443 | 10 |
| **MAD** | 0.9993024 | 5 | 0.9837090 | 12 | 0.9986686 | 5 | 0.9761528 | 8 |
| **AUPRC** | 0.9941390 | 15 | **0.9990008** | **1** | 0.9848855 | 14 | 0.9732409 | 11 |
| **Inverse AUPRC** | **0.9996026** | **1** | 0.9753107 | 14 | 0.9935138 | 9 | 0.9593002 | 15 |

Table 23: Detailed results of our AUC evaluation on Places365 and CUB.

## H  OLD SUMMARY TABLES

In this section we show our summary tables from the original submission for easy comparison during rebuttal. Will be removed from final version.

| | Missing Labels | | Extra Labels | | Pass |
|---|---|---|---|---|---|
| | Score Diff | Decrease Acc | Score Diff | Decrease Acc | |
| **Recall** | -0.2800 | 98.39% | 0.0010 | 0.00% | × |
| **Precision** | -0.0004 | 48.77% | -0.2727 | 99.96% | × |
| **F1-Score** | -0.1706 | 97.99% | -0.1518 | 99.94% | ✓ |
| **IoU** | -0.1765 | 97.99% | -0.1620 | 99.94% | ✓ |
| **Accuracy** | 0.0011 | 65.98% | -0.0041 | 99.96% | × |
| **Balanced Accuracy** | -0.1426 | 98.27% | -0.0016 | 86.78% | × |
| **Inverse Balanced Acc.** | -0.0003 | 51.50% | -0.0864 | 99.96% | × |
| **AUC** | -0.1396 | 92.03% | -0.0018 | 69.77% | × |
| **Inverse AUC** | -0.0015 | 69.08% | -0.2360 | 99.96% | × |
| **Correlation** | -0.0962 | 99.96% | -0.0951 | 99.96% | ✓ |
| **Correlation (top-and-random)** | -0.1136 | 90.29% | -0.0029 | 51.78% | × |
| **Spearman Correlation** | -0.0050 | 68.05% | -0.0013 | 52.63% | × |
| **Spearman Correlation (top-and-random)** | -0.1113 | 79.94% | -0.0013 | 50.99% | × |
| **Cosine** | -0.1538 | 100.00% | -0.1372 | 99.14% | ✓ |
| **Cosine cubed** | -0.1074 | 99.34% | -0.1047 | 99.96% | ✓ |
| **WPMI** | -0.1808 | 94.26% | -0.0296 | 99.96% | ✓ |
| **MAD** | -0.0016 | 52.63% | -0.1502 | 99.85% | × |

Table 24: Old: Combined results of our missing labels and extra labels test. We can see most evaluation metrics fail at least one of the tests.

| | Average Rank | | |
|---|---|---|---|
| **Method** | **Argmax Generation** | **AUC** | **Average** |
| **Recall** | 9.5 | 11.25 | 10.375 |
| **Precision** | 9.5 | 9 | 9.25 |
| **F1-score/IoU** | 6.25 | 9 | 7.625 |
| **Accuracy** | 9 | 8 | 8.5 |
| **Balanced Accuracy** | 7.75 | 7.5 | 7.625 |
| **Inverse Balanced Accuracy** | 8.5 | 8.75 | 8.625 |
| **AUC** | 7.75 | 9.75 | 8.75 |
| **Inverse AUC** | 11.75 | 11.75 | 11.75 |
| **Correlation** | 1.75 | 2.25 | 2 |
| **Correlation (top-and-random)** | 11.25 | 7.75 | 9.5 |
| **Spearman Correlation** | 16 | 16 | 16 |
| **Spearman Correlation (top-and-random)** | 14.75 | 12.5 | 13.625 |
| **Cosine** | **1.5** | **1.5** | **1.5** |
| **Cosine cubed** | 2.75 | 2.5 | 2.625 |
| **WPMI** | 4.25 | 6 | 5.125 |
| **MAD** | 7.25 | 6.5 | 6.875 |

Table 25: Old: Comparison of different evaluation metrics. Lower rank means better performance.

