# OpenReview forum: "A Principled Evaluation Framework for Neuron Explanations"
_ICLR.cc/2025/Conference — Submitted to ICLR 2025_

### Official Review · Reviewer_ibwF · 2024-10-31

**Soundness:** 2
**Presentation:** 3
**Contribution:** 2
**Rating:** 5
**Confidence:** 2

**Summary:**

This paper unifies various metrics used for evaluating mechanistic interpretability tasks into a single unified framework. It then introduces a series of "sanity checks" to assess the reliability of these metrics, demonstrating how some existing metrics fail these tests. Through experiments where the "ground truth" of a concept is known, the authors conclude that the "correlation" metric with uniform sampling yields the most accurate results.

**Strengths:**

1. A unifying framework for various metrics appears to be beneficial for researchers in this field.

2. The "sanity checks" conducted could provide valuable insights for those studying the topic.

3. The paper is thoughtfully and clearly written.

4. The paper combines both a "formal" aspect that unifies the definitions and practical experiments.

**Weaknesses:**

1. The primary contribution of this paper is in unifying various metric frameworks within mechanistic interpretability. However, this approach appears relatively "straightforward" and doesn’t seem to involve any technically complex process (the authors may clarify if they disagree). In traditional XAI, for instance, numerous papers propose unifying different notations, yet these often entail either technical or mathematical complexities ("demonstrating that X fits within this category is technically challenging because...") or yield unexpected insights ("it's surprising that X can also be captured under our framework because..."). This framework does not seem to meet those criteria.

2. I'm not sure why the "sanity checks" are regarded as a reliable evaluation framework for the metrics. Why should we trust the assurances provided by these "sanity checks" but not the guarantees offered by the metrics themselves?

3. The choice of the "correlation" metric appears somewhat rigid. Are there any potential issues with using this metric, even though it performed best in the experiments section?

**Questions:**

1. Are there any technical or conceptual challenges in constructing the underlying uniform framework? Does it reveal any non-trivial or surprising relationships, such as the unexpected categorization of certain metrics within this framework?

2. See other comments in the weaknesses section

3. It would be helpful if the authors could discuss the limitations of their unifying framework, the sanity checks performed, or aspects of the experiments section. Could the authors elaborate on some of the constraints or limitations of their work?

---

> ### Author Response · Authors · 2024-11-25
> **Author's Rebuttal Response**
>
> Thank you for your feedback! We have greatly strengthened our paper with additional experimental and theoretical results, please see **General response** for an overview.
>
> Below we respond to your concerns:
>
> **Weakness #1/Q#1: Benefits of unified framework**
>
> - We think a framework does not need to be complex to be useful, in fact to the contrary, we think the best frameworks feel simple and intuitive after you read them (but not before you read it).
> - We believe that the main contribution of our framework is that it provides conceptual clarity to the topic of neuron descriptions which has been lacking in existing work, and makes it clear how standard statistical metrics apply in this setting.
> - This framework allows us to analyze separate parts of each study, instead of considering them as a whole. For example, looking at a study in terms of metric, granularity, concept source etc. allows us to analyze existing studies with more detail (which was not done clearly in the literature before our paper; this is why we provide Table 1 in the main text Section 4, and the expanded version in Appendix E, Table 10), and combine good parts of different studies into new study designs. For example, evaluating F1-score by measuring recall in a human study and precision with a generative model is a promising study that would not be thought of without our framework.
> - Our framework also gives us several new insights such as:
>     - The fact that we can unify so many very different studies under one framework is surprising in itself. Before, they were thought to be more different than this, as can be seen in Table 1 (Section 4) and Table 10 (Appendix E) in the updated draft
>     - Existing papers sometimes use a different framing in terms of whether concept $c_t$(simulation) or neuron activation $a_k$(classification) is the predicted variable, as we discuss in Appendix A.2. This causes some confusion in terms, because the simulation framing of the Recall metric is equivalent to the classification framing of Precision.
>     - Methods that use generative models as the source for concept labels $c_t$ typically use metrics that fail the missing labels test as discussed in Appendix A.4. This actually might be necessary as labels from generative models typically are missing labels. This leads us to recommend combining these analyses with a different kind of evaluation such as a human study of recall.
>   - Separating metric from other details of the study allows us to focus on analyzing metric choice alone, and inspired by the framework we develop sanity checks to find bad metrics.
>
> **Weakness #2: Why trust sanity checks over metric outputs themselves**
>
> Thank you for the response. We believe you would find our new theoretical sanity check in Appendix B interesting. In that section, we find that metrics failing the sanity checks is mostly caused by poor performance on imbalanced data. We cannot just trust the guarantee offered by the metric at face value every time. For example, if someone claims their model has 99.9% accuracy on a binary classification task, does that mean they have a good model? Not necessarily, they could just have a dataset where only 0.1% of the inputs are positive and the model simply predicts no every time. This is precisely why accuracy is not a good metric for evaluating neuron description accuracy, as the concepts of interest are often highly unbalanced, and in response we can see accuracy fails our sanity checks in Table 2 and Appendix B. The point of our sanity checks is to identify metrics whose output scores are not always indicative of good predictions, in particular on imbalanced data.
>
> **Weakness #3: Limitations of correlation**
>
> Thank you for the suggestion. Our proposed sanity check tests can be regarded as providing a necessary condition for good evaluation metrics. Thus, a good metric should pass the tests and for those metrics that do not pass the tests, they should not be used. As a result, we have toned down our language regarding correlation as the best metric, and updated the conclusions in Sec 6 to say that we focus on identifying a set of good metrics to be more in line with the strength of our experimental results. We have not identified any specific issues with correlation but some may exist.
>
> **Question #3: Limitations:**
>
> Thank you for the suggestion. We have added a comprehensive discussion of the limitations of our framework and experimental results in Appendix A.1.
>
> We believe we have addressed all your concerns, please let us know if you have any remaining questions or concerns, and we would be happy to discuss further!

---

> ### Author Response · Authors · 2024-12-02
> **Request for Feedback**
>
> Dear Reviewer ibwF,
>
> Thanks again for reviewing our paper! With the discussion period ending today, we wanted to follow up as we still haven’t heard from you. We have conducted extensive additional experiments and believe we have resolved your concerns but please let us know if that is not the case. We would appreciate your feedback or an acknowledgement that you have read through rebuttal response.

---

### Official Review · Reviewer_DS3i · 2024-11-02

**Soundness:** 2
**Presentation:** 2
**Contribution:** 3
**Rating:** 6
**Confidence:** 4

**Summary:**

The paper studies the problem of evaluating neuron explanations. Specifically, it proposes a unification of some of the metrics used for neuron explanations and two sanity checks that a good metric needs to pass. The sanity checks work by replacing the class labels associated with the data points in the probing dataset. While the first check removes half of the labels for a given class, the second doubles their count. The idea is to use such labels as a concept, keep the explanations associated with the neurons the same, and check the sensitivity of the metrics to the change. To compute the ground truth explanations (which will be kept fixed), the authors consider the output neurons of vision models where the ground truth is the label to be predicted and the class that optimizes the overlap between firing rate and class samples for a hidden layer. The authors claim that good metrics should exhibit a large change in the scores and, based on such claims, identify and claim the best metric among the tested ones as the one where the gap is the largest.

**EDIT AFTER REBUTTAL**: Given the current changes to the manuscript and the promised ones (please refer to my Summary of Review After Rebuttal and the authors' response for a detailed explanation), most of my concerns have been or will be addressed. The only remaining issue is related to the significance of the tests in Section 5. However, considering the improvements in all other areas, I am leaning towards acceptance.

**Strengths:**

- Evaluating the quality of explanations and addressing the fragmentation of the metrics is arguably the most important problem in the field of explainable artificial intelligence. The field needs more papers in this direction.

- The development of more sanity checks for explanations is important and would be a good contribution


- The pure results in the tables (without the analysis and claims - see weaknesses and questions), are significant and important to the field and can raise awareness about the weaknesses of metrics used for evaluating explanations.

**Weaknesses:**

- The most significant weakness relates to the claims and analysis of the results. **There is a mismatch between what the authors claim,** (assess which metric is the best one to evaluate explanations) **and what the results and the sanity check support** (asses which metric is the most sensitive to the concept label noise in a probing dataset -  i.e., the (non-) robustness of metrics in the presence of concept label noise). In a few words, claiming that the best metric to evaluate explanations is one that exhibits the largest sensitivity to label noise is misleading and not conceptually precise. I would suggest maintaining all the experiments and changing the narrative and the analysis to better reflect the results of the experiments (for example moving the focus from "finding the best metric for evaluating neuron explanations" to "benchmarking metrics in the presence of label noise").


This first point requires a detailed description, provided below. The authors are encouraged to provide further clarification or supporting evidence or adapt the narrative and claims of the paper to the raised concern:

Consider the setup of the output neurons considered by the authors. In this case, the tests:
- do not change the input (so the concepts are still included or not included in the input),
- do not change the explanation (which is truthful as claimed by the authors to be the ground truth), and
- do not change the neuron’s behavior.

This setup means that the explanation is still the right one because it reflects what the neuron actually learned and its behavior. Therefore, in an ideal (and unrealistic) scenario, a perfect metric should still assign a high score with the explanation associated with the neuron both before and after the tests based on the semantics of the input and the behavior of the neuron and ignoring the label associated with the input.

Conversely, *what the tests are doing is **introducing label noise** into the probing dataset.*
The problem of label noise is well-studied and well-known in binary classification setups. Ideally, a perfect metric should be immune to label noise, as stated before. However, such a metric is unrealistic, and at the practical level, f1 or AUC are the best and standard de facto metric. Analyzing the results in the table, the best metrics identified by the authors are more sensitive to label noise than these standard metrics for binary classification problems. It is not clear why a metric should be more sensitive to label noise than these latter metrics. This is a good sign that the claim of the authors about the criteria to establish the best metric could be misleading and needs to be reconsidered or further assessed. Again, there is nothing wrong with the experiments or the sanity checks; they could be a wonderful resource against adversarial attacks or to check robustness against label noise. **However, I argue these tests should not be used to identify the best metric to evaluate the quality of explanations.**

- **There is a mismatch between the generality of the claims (best metrics for neural explanations) and the specific settings analyzed**. Given the scope and the expected impact of the paper, it is essential to be rigorous and to state explicitly at the beginning of the paper (alongside the claims) what are the assumptions and settings where this behavior has been observed and holds true. Currently, these must be inferred implicitly from the paper. In the current state, the experiments focus only on (i) the vision domain (ii), models trained on ImageNet, (iii) settings where the concepts correspond to the classes on which the model was trained on, and (iv) concepts are identified by their binary presence. Several of these assumptions are not standards (e.g., (iii) (iv)). The authors should provide more evidence of the generality of their claims in different domains (e.g., nlp), models (trained on different datasets and not trained on the concepts), and setups (e.g., non-binary concepts) claims and explicitly state these assumptions.

- The organization of the paper could be slightly improved. For example, the metrics evaluated in this paper are presented only as equations in the main text. There is no textual description of them. Since they are the main focus of the paper, they deserve more attention.  It is unclear what these metrics measure, why they were proposed, and what their strengths and weaknesses are. Some details, such as the setup where they were proposed (e.g., domain, granularity) and ranges, are included in tables and the appendix, but these are not sufficient to fully understand the metrics. To save space, certain sections that are not the main focus of the paper do not need detailed descriptions in the main text (e.g., function 2 in Section 2.3 and associated discussions, or even the generation part of Section 2.2).

- It is unclear what rationale and process were used to select the metrics analyzed. As with the previous points, since the claims involve identifying the best metric to use in future work, it is important to adopt a clear rationale and adhere to it.  Some metrics included in the analysis have been used only in one paper (or at least it seems so by looking at the table references), so popularity does not seem to be the criterion. Meanwhile, metrics used in other papers are missing from the analysis. Adopting a rigorous strategy (e.g. popularity, publication venues, date of publication, compatibility with the proposed unification, etc.) and explicitly stating the procedure used to select the metrics would strengthen the paper.

**Questions:**

All the weaknesses listed above are indirect questions for the authors. Therefore, please consider them as points to discuss. The score should be considered a placeholder. I will significantly revise it if the authors provide further clarification or supporting evidence or adapt the narrative and claims of the paper to the raised concerns.

All the following questions did not affect my evaluation but I would appreciate a clarification:
- What is the rationale for assigning citations to the metrics in Table 1? Some of the metrics have been used in multiple papers, but only one of them is reported (e.g. WPMI)
- There are some simplifications in the setup and definitions of the tested metrics and the way these metrics have been used in the original papers (e.g., correlation score or IoU). Is this correct or are they the same? If the answer is negative, is there a mathematical proof ensuring that the results for the simplified and unified versions hold for the settings tested by the original papers? If so, please include this information in the appendix.
- It is not clear why some metrics, like the Cosine cubed, are included. They have been used only as an optimization metric for CBMs and not strictly for comparing and evaluating neuron explanations (even if the target is related). If the authors prefer to include this kind of metrics, there are many other similar optimization metrics in the CBM research area that should be included.
- The motivation for considering correlation a better choice than cosine similarity seems quite weak as reported in the paper (just one sentence vs better results overall!). Can authors provide an example of “significant errors when explaining neurons whose average activation is different from zero”?

---

> ### Author Response · Authors · 2024-11-25
> **Author's Rebuttal Response 1/3**
>
> Dear Reviewer DS3i,
>
> Thank you for the detailed feedback and thoughtful comments. We have added extensive additional experiments in response to reviewer feedback, please see the General response for a summary. In particular, we think our theoretical missing/extra labels test in **Appendix B** (in the revised draft) will be of interest to you and potentially addresses many of your concerns, so we hope you could take the time to read through our results and please let us know if you have any additional feedback.
>
> In response to your specific concerns:
>
> **Weakness #1: Framing missing/extra labels test as label noise**
>
> You raise an interesting point, and we agree that one way to view these tests is in terms of label noise. However, we disagree with your conclusions, in particular that a good metric should be robust to label noise. This is based on the following reasons:
> - To clarify, we introduce a very large amount of label noise, i.e. in extra labels tests, 50% of the positive labels are now incorrect. While robustness to low amounts of label noise could be desirable, if a metrics output does not change in response to a very large amount of noise, it is insensitive/not measuring the right thing. This can be seen in other settings, for example in the context of adversarial learning, robustness to small perturbations is desirable, but the only way to be robust to very large perturbations is a constant classifier that predicts the same class regardless of input, which is clearly not desirable. Another example of this same phenomenon can be observed in the Sanity Checks for Saliency Maps [R1], which inspired our sanity checks. In their Figure 2 they perform a randomization test on network parameters. While you might initially think that robustness to small changes in NN parameters is desirable for a saliency map method, if we completely randomize the network weights and the results still don’t change it means the saliency map is insensitive to network parameters in problematic ways instead of being robust.
> - As an extreme case, consider adding uniform random noise with 50% chance of flipping each label. Since there is little to no information left in the labels, clearly the evaluation metric should output a different score.
> - We do not think a good evaluation metric should be robust to label noise. A good learning algorithm should be, but an evaluation metric should not. For example, if you measure the accuracy of a good classifier, but 10% of your labels are incorrect, you should get a lower accuracy.
> - Our setting is a little different from standard label noise settings, where typically every label has a uniform probability of being incorrect. Instead, our label noise completely depends on the ground truth label, i.e. in missing labels test the labels where concept is present have 50% chance of being incorrect, while labels where concept is not present are always correct.
>
> Overall we think that because we introduce a very large amount of label noise, metric not changing its value in response is not a case of positive robustness, but instead reveals a problematic insensitivity to certain features of the labels.
>
> Additionally, in Appendix B, we represent an alternative way to view this test, which we think can provide more theoretical insights to our proposed tests. In particular, in an ideal setting where a neuron’s activations perfectly align with a concept, i.e. $a_k = c_{t_k}$, our Missing and Extra Labels tests correspond to being able to differentiate between the following 3 concepts:
> - $c_{t_k}$: The perfect predictor for neuron $a_k$, with precision=1 and recall=1.
> - $c_{t_k}^{-}$: (Missing labels) This concept has Precision=1 since whenever a concept is present, the neuron is also active, and Recall of 0.5 since only half to inputs where the neuron activates now have the concept.
> - $c_{t_k}^{+}$: (Extra labels) Inverse from above, this concept has Precision=0.5 and Recall=1.
>
> Clearly, the concept with perfect recall and precision should achieve a higher score on the evaluation. However, this is not the case for most metrics as we show in Appendix B, as they fail one or both of these tests, especially if data is imbalanced, i.e. $c_{t_k}$ is only rarely positive. This motivates our test, a good metric should be able to differentiate these concepts regardless of concept imbalance, and the passing metrics such as F1-score/IoU and Correlation do.

---

> > ### Author Response · Authors · 2024-11-25
> > **Authors Rebuttal Response 2/3**
> >
> > **Weakness #2: Sanity check should not determine best metric**
> >
> > Indeed this is not how we intend to use the missing/extra labels test. Instead, the Missing/Extra labels tests are intended to be used as a pass/fail test to find bad metrics that are problematically insensitive to precision or recall. In practice, this means that all metrics that receive two checkmarks in Table 2 are deemed good, while the rest fail. Beyond that, we do not differentiate metrics based on results in Table 2.
> >
> > To further evaluate metrics, we rely on the completely different evaluation strategies in Section 5 and Table 3 that measure how well the metrics do on neurons where we know their ground truth concepts which helps us determine the “best” metric. Overall, we don't think these findings are as strong as our sanity check, and in response we have edited the writing to reduce focus on finding the “best” metric and focus instead on finding a set of “good” neurons.
> >
> > **Weakness #3: Claims more general than results**
> >
> > Thank you for the feedback. In response, we have greatly expanded the experimental setup of our results to two additional datasets CUB200 and Places365, where on the CUB dataset we study concepts that represent lower-level features like wing-color instead of the final classes the model was trained on.
> >
> > In addition, our theoretical results on missing/extra labels test described in Appendix B are not tied to any domain, dataset or concept granularity, giving us confidence that these sanity check results are a fundamental feature of the evaluation metric instead of any specific to anything in our experimental setup.
> >
> > **Weakness #4: Organization, lacking description of metrics**
> >
> > Thank you for the feedback, in response, we have added a short textual description to each metric we study. To address the space issues caused by this and feedback from other reviewers that metric descriptions are taking too much space, we have moved some of the less important metric definitions to Appendix C, while keeping some of the main metrics in main text.
> >
> > **Weakness #5: Selection criteria for metrics**
> >
> > In general we tried to be as inclusive as possible, and include all the metrics we are aware of being used to **evaluate** neuron descriptions in the literature, as well as any standard statistics metrics that have not been used for this task but we think might perform well/be interesting to study. It is likely that we have missed some interesting metrics in related work, if you can provide any such references we would be happy to include these metrics in the future.
> >
> > **Question #1: Citations in Table 1**
> >
> > We tried to cite every paper that uses this metric to **evaluate** neuron descriptions.  For example, WPMI you mention was also used by MILAN[R2]. However, in MILAN, WPMI was not used to evaluate the quality of generated descriptions, but instead only used it to guide the explanation generation. Their evaluation in MILAN is instead focused on comparing textual similarity between the automatically generated neuron description and descriptions generated by MTurk workers. This evaluation is not included in Table 1 as it does not fit within our framework, but we do not think this type of approach is a reliable way to evaluate neuron description quality as it assumes that a single perfect description exists for each neuron (and that we can find it via MTurk). We have added discussion on this in Limitations of our framework in Appendix A. In contrast, the authors of CLIP-Dissect analyze using the (soft)WPMI score as a metric of evaluating how good the generated explanations are in their Appendix A.8, which is the part of the paper referenced in that line of Table1.
> >
> > **Question #2: Simplifications of metrics**
> >
> > Our definitions for correlation score and IoU are mathematically exactly the same as the metrics used in the papers cited in Table 1. The metrics marked with “~” are slightly simplified in our framework, such as Recall, and we discuss the differences to original evaluation in Appendix A.1.
> >
> > **Question #3: Why is cosine cubed included**
> >
> > We have included cosine-cubed as in the original LF-CBM paper [R3], in addition to optimization, it is also used to evaluate the descriptions in a subtle way. In particular, the authors said in their paper: “Finally to make sure our concepts are truthful, we drop all concepts j with sim(tj,qj) < 0.45 on validation data after training Wc.”, i.e. they measure cosine similarity on validation data after optimization to determine which concepts were learned well, which is a type of evaluation. However, it is not very important to our study and we can remove the results if you think that would be more consistent.

---

> > > ### Author Response · Authors · 2024-11-25
> > > **Author's rebuttal response 3/3**
> > >
> > > **Question #4: Motivation for correlation over cosine is weak**
> > >
> > > Thank you for pointing this out. To further showcase the failure case of cosine similarity, we have added an experiment in Appendix F.2 (Table 13) in the revised draft, showing that adding a constant to the neuron activations reduced argmax generation accuracy of cosine from 99.07% to 0.93% on CUB concept neurons, while all other metrics remain unchanged. In addition, after adding new experimental setups, we found that Correlation now actually performs the best on average in Table 3 (but the two perform very similarly).
> > >
> > > In summary we have conducted greatly extended experiments showing the robustness of our results to new settings, as well added theoretical justifications, as well as discussed in depth why we think robustness to label noise is not a helpful way to view our sanity check results. We hope we have addressed your concerns, but let us know if you have any remaining.
> > >
> > > **Reference:**
> > >
> > > [R1] Adebayo, Julius, et al. "Sanity checks for saliency maps." Advances in neural information processing systems 31 (2018).
> > >
> > > [R2] Hernandez, Evan, et al. "Natural language descriptions of deep visual features." International Conference on Learning Representations. 2022.
> > >
> > > [R3] Oikarinen, Tuomas, et al. “Label-free concept bottleneck models.” International Conference on Learning Representations. 2023.

---

> ### Comment · Reviewer_DS3i · 2024-11-26
> **Response to Weakness #1 Reply**
>
> I would like to thank the authors for their detailed response. While I plan to take additional time to analyze the full response, I am prioritizing speed over precision in this reply to allow the authors as much time as possible to address the remaining concerns.
>
> One of my earlier concerns has not yet been fully resolved. I agree with the authors that, at a practical level, a metric should be sensitive to label noise. However, as highlighted in my original review, it remains unclear **why a "suggested/better" metric should be more sensitive to label noise than the F1-Score**, for instance. The F1-Score is widely recognized as the standard metric for binary classification in the presence of label noise (in the general case, without specific constraints).
>
> Could the authors elaborate further on this point?
>
> Specifically, should the objective of this research area be to identify the most sensitive metric to label noise?
> In this context, could the authors provide examples demonstrating cases where the F1-Score is a better metric for binary classification but not for neural explanations, and conversely, where correlation is superior for neural explanations but less effective for binary classification?
> Alternatively, it would be helpful if the authors could provide a reference supporting the claim that correlation is a better metric than the F1-Score for binary classification problems, or offer a theoretical proof or justification.
>
> Thank you in advance for your clarification.

---

> > ### Author Response · Authors · 2024-11-26
> > **Reply to response**
> >
> > Thank you for the quick reply to allow for more discussion!
> >
> > We would like to clarify that we don’t think a metric should be more sensitive to label noise than F1-score(see line 429 in the updated manuscript). F1-score is a good example of a metric that exhibits correct levels of sensitivity to this kind of label noise, and we think good metrics should exhibit comparable amounts of sensitivity to this noise as F1-score does. For selecting the best metric, we do not care about sensitivity beyond a certain point, and both F1-score and correlation pass our sensitivity tests on Table 2 so we do not further differentiate between them based on these results. Please refer to our response to Weakness #2 as it is more focused on this point.
> >
> > Our findings suggesting that correlation is a better metric for neuron descriptions are instead based on different experiments in Section 5 and Table 3, where we compare whether these metrics assign the higher scores to “ground truth” concepts than other explanations on neurons where we have access to such ground truth, for example final layer neurons. We experimentally find that correlation performs better on this setting with real neurons than F1-score does. We think this is mostly because real neurons do not have binary activations, but to use binary metrics like F1-score we have to binarize them(and find the right parameter $\alpha$), and this process loses some information which leads to decreased performance.
> >
> > Overall we think F1-score is likely the better choice for binary classification problems, but since neuron activations are not binary, correlation often performs better on neuron explanations, as we show in Section 5.
> >
> > We have also changed the language in our manuscript to focus less on identifying the best metric and more on identifying a set of good metrics, which includes F1-score/IoU. In the revised manuscript, we have included this discussion in Sec 6 Conclusions, and we are copying below for your reference:
> > “Other good metrics include cosine cubed and AUPRC. F1-score or IoU can also be a good choice, but requires choosing an activation cutoff α and it is unclear how to best make this choice.”
> >
> > Please let us know if you still have any remaining concerns and we would be happy to discuss further!

---

> > > ### Comment · Reviewer_DS3i · 2024-11-28
> > > **Clarifications on the significance of experiments in 5.1**
> > >
> > > Thank you to the authors for their clarification. I will update my score and provide more detailed feedback before the end of the discussion period. As with my previous response, I am prioritizing speed over completeness in this reply.
> > >
> > > Could the authors please clarify the connection between the experiments in Section 5.1 (argmax generation) and the assessment of a metric's quality as an evaluation metric? This relationship remains unclear to me in the current version of the paper. To improve clarity and ease of understanding, I suggest that the authors include a description of the experimental setup along with a brief introduction to these experiments in the camera-ready version.
> > >
> > > From my understanding, the phrasing in the paper, and the previous authors’ response, in the argmax generation experiment, the metric appears to be used simultaneously for both optimization and evaluation. If this interpretation is correct, the results seem to suggest that correlation is a better metric as an optimization metric rather than as an evaluation metric (with the caveat that we would expect to use distinct metrics for optimization and evaluation for a fair comparison). Are the authors leaning towards correlation because it passes the sanity checks as an evaluation metric and also performs better as an optimization metric in the scenarios considered? Or is there additional significance tied to the experiment (or meaning/setup) in Section 5.1 that has not been explicitly highlighted?

---

> > > > ### Author Response · Authors · 2024-12-01
> > > > **Clarification on section 5.1**
> > > >
> > > > Thanks again for the quick response!
> > > >
> > > > Regarding section 5.1, we study argmax generation as it is an interesting connection between explanation generation and evaluation, as we discuss in **Section 2.2 (III) Connection between Evaluation and Generation:**. In particular, we measure whether the correct concept gains the highest evaluation score for each neuron, which is a desirable property for a good metric.
> > > >
> > > > In a sense, we only use each metric for explanation generation in section 5.1, and evaluate the generated concepts by whether they match the known ground truth for that neuron (this is not a feasible evaluation method for hidden layer neurons as we do not know the ground truth).
> > > >
> > > > In a real-world setting with argmax generation, you would use argmax generation to describe the neurons using a training split of your probing data, and then use the same or a different metric on a test split(of probing data) to evaluate how good said descriptions are to avoid overfitting.
> > > >
> > > >
> > > > On the other hand, in section 5.2 we perform a different experiment on final layer neurons that directly measures the evaluation performance of different metrics. We run each evaluation metric on all (neuron, concept) pairs and then measure whether the metric consistently assigns higher scores to correct pairs (concept = gt concept for neuron) than it does to incorrect pairs. A good metric should cleanly separate the scores of the correct pairs from the scores of the incorrect metrics, and we measure this with AUC (note this is meta-evaluation AUC, different from AUC defined in section 3.1 with different inputs).
> > > >
> > > > We will clarify these connections and the motivation for section 5.1 in the final version of the paper. Overall we lean towards correlation as it passes the sanity checks, performs better on the generation test in section 5.1 and the evaluation test in section 5.2.
> > > >
> > > > Please let us know if you have any further questions.

---

### Official Review · Reviewer_y8T5 · 2024-11-02

**Soundness:** 3
**Presentation:** 4
**Contribution:** 3
**Rating:** 6
**Confidence:** 4

**Summary:**

The paper addresses a critical issue in neuron explanation methods: the works of neuron explanation methods often use completely different metrics to evaluate how good their explanations are. It proposes a unified mathematical framework to unify existing evaluation metrices.

Furthermore, the paper introduces two sanity checks for evaluation metrics: the Missing Labels test and the Extra Labels test. These tests are designed to assess the robustness of commonly used metrics. The findings reveal that many of these metrics fail to pass at least one of these fundamental tests. A key outcome of the research is the identification of Correlation as the most effective overall evaluation metric.

**Strengths:**

1. The study tackles an important research question in XAI by aiming to compare and contrast existing evaluation metrics for neuron explanation methods and understand the evaluation pipeline with increased clarity.

2. The definitions and explanations related to the mathematical framework are well articulated.

3. The introduction of two sanity checks based on the mathematical framework seems sensible. These checks provide practical tools for assessing the robustness of the evaluation metrics.

4. The findings effectively highlight the shortcomings of existing evaluation methods and underscore crucial considerations for designing robust evaluation metrics. This enhances our understanding of how to measure the effectiveness of neuron explanations accurately.

**Weaknesses:**

1.	Ambiguity in Parameter Selection: The paper does not provide a clear rationale for the choice of threshold hyperparameters in equations (5) and (6), nor for the value of \alpha in the second experimental setting, or the ratios 0.5/2 in the first and second sanity checks. It remains unclear how these values were determined and whether different values might impact the results.

2.	While I understand that the paper focuses on evaluating Function 1, considering the title “A Principled Evaluation Framework for Neuron Explanations,” it would be beneficial to discuss the relationship between evaluating Function 1 and Function 2 more thoroughly. It would be advantageous to demonstrate whether the current evaluation framework could be adapted to evaluate Function 2 with some modifications. If this is not feasible, I would suggest that the authors narrow their claims prior to Section 2.3 to better reflect the specific focus of the framework.

3.	In Setup 4 of Table 7, all methods fail to achieve high accuracy. Could the authors explain why this is the case or why this setup is particularly more challenging? In this scenario, even the best method achieved less than 80% accuracy. Does this indicate that the current best evaluation methods are still not perfect?

**Questions:**

See weaknesses.

---

> ### Author Response · Authors · 2024-11-25
> **Author's Rebuttal Response**
>
> Dear Reviewer y8T5,
>
> Thank you for the detailed review and positive feedback! We have conducted extensive additional experiments, based on all reviewers feedback, please see the **General response** for an overview of all new results.
>
> Below we would like to address specific concerns you have:
>
> **Weakness 1a: Parameter selection:**
>
> - As described in Line 511 in the original manuscript (Line 482 in the updated draft): For all experiments we split a random 5% of the neurons into a validation set. For metrics that require hyperparameters such as $\alpha$, we use the hyperparameters that performed the best on the validation split for each setting. We then report performance on the remaining 95% of neurons. Note that this is how we chose $\alpha$ for results in Table 3.
>
> - For Missing/extra labels test on final layer neurons, we used the $\alpha$ that gave the best performance in terms of AUC evaluation. For hidden layer neurons we used $\alpha = 0.005$ for all metrics following Network Dissection [1]. We have clarified this with additional explanation in Section 3.1 and Appendix G.
>
> - The threshold parameter $b_{\alpha}$ in Eq. 5 is determined by the value of $\alpha$ and is not an independent parameter.
>
> - For Eq. 6, we used 0.5 as this is the conventional cutoff when measuring accuracy in binary classification and it seems hard to justify using a different cutoff for a probability.
>
> **Weakness 1b: Ratio selection in Missing/Extra labels test**
>
> Thank you for the suggestion, we have conducted an ablation using different ratios in Appendix F.1, please see Tables 11 and 12. We find that our results are not sensitive to this choice of ratio, and instead we get very consistent results in terms of which metrics pass/don't pass the tests.
>
> **Weakness 2: Missing Function 2**
>
> Thank you for the feedback. We have added the following discussion under Limitations in Appendix A.1 to make it clear that this work we focus on evaluating function 1 only.
>
> *“Function 2: Activation → Output: Probably the most significant limitation of our framework is that it is focused on evaluating function 1 only (Input → Unit Activation) as discussed in Section 2.3, and we believe measure Function 2 is equally important. While currently our framework is meant for function 1 only, we believe many of the ideas and metrics we discussed could be useful in evaluating function 2. For example, in a generative model we could use the same metrics to measure similarity between unit activation and the presence of a specific concept in the output. However in function 2 there are additional considerations and things like measuring difference in outputs when changing the unit activation are likely more important. We believe extending this framework or creating a similar one for function 2 evaluations is an important direction for future work.”*
>
> **Weakness 3: All methods have low accuracy in Setup 4 of Table 7**
>
> This low performance is caused by limitations in our pseudo-labels. Specifically, the pseudo-labels from SigLIP are quite accurate for original imagenet classes, but become much less accurate for superclasses, making it impossible to reach great accuracy on Setup 4 (while Setup 2 many methods perform well). Note old table 7 is now in Table 20 of the revised manuscript
>
> We hope we have addressed all your concerns, please let us know if you have any additional questions and we would be happy to discuss further!

---

> ### Author Response · Authors · 2024-12-02
> **Request for Feedback**
>
> Dear Reviewer y8T5,
>
> Thanks again for reviewing our paper! With the discussion period ending today, we wanted to follow up as we still haven’t heard from you. We have conducted extensive additional experiments and believe we have resolved your concerns but please let us know if that is not the case. We would appreciate your feedback or an acknowledgement that you have read through rebuttal response.

---

### Official Review · Reviewer_1n3c · 2024-11-02

**Soundness:** 2
**Presentation:** 3
**Contribution:** 2
**Rating:** 3
**Confidence:** 5

**Summary:**

This paper focuses on the evaluation of explanation methods for neurons in DNNs. The authors first formalize the evaluation task in a unified framework. Then, they identify the theoretical flaws of existing evaluation metrics, and further propose two sanity checks for evaluation methods. In experiments, they find that most evaluation methods cannot pass the sanity check, while the correlation metric and cosine similarity exhibited good performance.

**Strengths:**

1. The authors focus on an important topic of the evaluation of explanation methods.
2. The authors compare many different evaluation metrics in experiments.

**Weaknesses:**

1. This paper is only limited to explanation methods based on simple concepts. Specifically, the explanation in this paper only refers to whether a neuron represents a specific concept, where the concept is simplified as the class label or superclass labels. However, the explanation for a neuron is much more complex and not limited to representing a single concept, such as the receptive filed of a neuron or saliency map. The proposed evaluation principles cannot be scaled to other explanation methods.
2. The proposed method is ad-hoc. Why do you only adjust the concept label for testing? Why not try to adjust neural activations? The theoretical guarantee for the completeness of the proposed method is needed.
3. The mathematical formulation of the explanation $t_k$ is unclear. Does it represent a scalar, a vector, or a text description?
4. All the metrics and sanity checks seem to be limited to the explanation for a neuron. How do you compute these evaluation metrics and sanity check if the explanation is based on linear combination of neurons? Many papers have claimed a concept is usually jointly represented by the combination of multiple neurons, rather than a single neuron.
5. The completeness of sanity checks in the paper is not guaranteed. In other words, even if an evaluation metric passes two sanity checks, it still cannot grauantee this metric to be a faithful metric. The limitation of the evaluation method needs to be formulated in mathematics.
6. All experiments in the paper are conducted using class labels or superclass labels as the concept. If the concepts are more fine-grained attributes like stripe or wings, is the proposed method powerful enough to still identify the difference between different evaluation metrics?

**Questions:**

Please refer to the weakness part.

---

> ### Author Response · Authors · 2024-11-25
> **Author's Rebuttal Response (1/2)**
>
> Dear Reviewer 1n3c,
>
> Thank you for the review! In response to feedback from you and other reviewers, we have greatly expanded our experimental results and restructured some of the paper, please see General response for description of all the changes and the updated manuscript.
>
> Below we clarify and respond to specific concerns you have:
>
> **Weakness 1: Limited to explanation based on simple concepts:**
>
> Our framework can evaluate any textual explanation, and is not limited to class/superclass labels at all. To showcase this, we have added experiments using the fine-grained concepts in the CUB200 dataset, such as \textit{black wing-color}. In addition, our framework can be easily extended to non-textual concepts such as a prototype-based explanation as long as we have a way to determine the concept vector $c_t$ for that concept as we discuss in Appendix A.1.
>
> One thing that is unclear to us is about your comments on the other methods such as a saliency map or the receptive fields. To clarify, in this work, we are focused on global neuron explanation, while a saliency map is a local input importance method which does not belong to neuron explanation methods. If you are aware of a method that produces these types of global explanations for a single neuron, please provide a reference and we would be happy to look into this.
> In short, our aim is not to evaluate every explainable AI method, but specifically global neuron descriptions, which provide global explanations to characterize neuron functionalities.
>
> **Weakness 2a: Why do you only adjust the concept label for testing? Why not try to adjust neural activations?:**
>
> May we ask if this question refers to the missing/extra labels test? To clarify, we focus on adjusting the concept as typically in neuron explanation the neuron is considered fixed and the concept is the thing we can change. In addition, the current formulation of the test relies on $c_t$ being binary, and neuron/unit activations are not binary in practice, so it’s not clear how to extend this. However, in our theoretical setting in Appendix B where we consider an ideal neuron with $a_k = c_{t_i}$, we can see that these two formulations give the same results.
>
> In particular, a missing labels test on neuron $a_k$ that activates on fraction $\gamma$ of the neurons is equivalent to an extra \textit{activations} test on neuron $a_s$ that activates on fraction $\gamma{}/2$ of the inputs. That is, $M(a_k, c_{t_k}^-) = M(a_s^+, c_{t_s})$ for any metric.
>
> **Weakness 2b:The theoretical guarantee for the completeness of the proposed method is needed.**
>
> Can you please clarify what you mean by this? What does completeness mean in this context of global neuron explanations?
>
> **Weakness 3: What is $t_k$**
>
> $t_k$ is a text description. We have clarified this in Section 2.2 (Line 101) in the revised draft.
>
>
> **Weakness 4: Missing linear combinations of neurons.**
>
> Computing the tests on a linear combination of neurons is mathematically no different from calculating the tests on a single neuron. In fact, every neuron in a feedforward network is a linear combination of previous layer neurons. In mathematical terms, let A be a d x n matrix of the activations of all d neurons in a layer on all inputs. A regular neuron’s activations $a_k = A_k$, i.e. the kth row of A. For a linear combination, we can represent its activation pattern as a virtual neuron $\hat{k}$ $a_{\hat{k}} = wA$, where $w$ represents the coefficients of the linear combination. To showcase this ability in practice, we trained a Linear probe for CUB concepts on top of CLIP image encoder embeddings, where each concept is represented by a linear combination of neurons and included the results in our evaluation. Please see Tables 19, 21 and 23 in Appendix G for detailed results.

---

> ### Author Response · Authors · 2024-11-25
> **Author's Rebuttal Response (2/2)**
>
> (continued for Reviewer 1n3c)
>
> **Weakness 5: Passing sanity checks does not guarantee the metric is faithful**
>
> Our approach is inspired by the sanity checks of Adebayo et al [R1], which have been very influential in the field of local feature importance/saliency maps. To clarify, passing the sanity checks proposed by Adebayo et al. [R1] does not guarantee that a method is good, but not passing them means that a method certainly is bad. In other words, one can think of this kind of sanity check as a necessary condition for a good metric. Our sanity checks are intended to be used the same way, to identify bad metrics and are not designed to guarantee a method passing them is good. We do not believe this reduces the value of having these tests and similar to the sanity checks of Adebayo et al. [R1], our checks can help guide the field in the right direction. We have clarified this point and included discussion of this limitation in Appendix A.1.
>
> **Weakness 6: Lacking fine-grained concept experiments**
>
> Thank you for the suggestion. Following your recommendation, we have conducted all our experiments additionally on the fine-grained concepts of CUB200 dataset, using two separate models, a standard Concept Bottleneck Model trained on this data, as well as a linear probe trained on CLIP embeddings. Please see Tables 18, 19, 21 and 23 in Appendix G for detailed results. In terms of final layer metrics our method can very clearly differentiate between different metrics, see Table 21 for example. For missing/extra labels test there is less difference between metrics, which we identify is caused by the CUB concepts being more balanced (positive on at least 5% of the inputs) than for example ImageNet classes (positive on 0.1% of the inputs). See Tables 4 and 5 in Appendix B for detailed study on the effect of concept balance on our sanity test results.
>
> **Reference:**
>
> [R1] Adebayo, Julius, et al. "Sanity checks for saliency maps." Advances in neural information processing systems 31 (2018).
>
> **Summary**
>
> In short, there is one comment (Weakness 2b) that we hope the reviewer could help to clarify, and we would be happy to address the reviewer's question or concerns. We believe we have addressed the reviewer's other concerns in weakness. Please let us know if you have any additional concerns and we would be happy to discuss further!

---

> ### Author Response · Authors · 2024-12-02
> **Request for Feedback**
>
> Dear Reviewer 1n3c,
>
> Thanks again for reviewing our paper! With the discussion period ending today, we wanted to follow up as we still haven’t heard from you. We have conducted extensive additional experiments and believe we have resolved your concerns but please let us know if that is not the case. We would appreciate your feedback or an acknowledgement that you have read through rebuttal response.

---

### Official Review · Reviewer_hRsh · 2024-11-03

**Soundness:** 3
**Presentation:** 3
**Contribution:** 3
**Rating:** 5
**Confidence:** 3

**Summary:**

The paper introduces a unified mathematical framework for evaluating neuron explanations and proposes two new sanity checks to assess the reliability of these evaluation metrics. The work highlights the shortcomings of many current metrics and recommends correlation (with uniform sampling) as the most reliable metric. Through theoretical analysis and comprehensive experiments, the paper provides insights into the biases of common metrics and emphasizes the importance of unbiased sampling.

**Strengths:**

This paper is well-written and adress an important problem to the field of explainability:

1. **Unified Framework**: The paper effectively consolidates various existing evaluation metrics under one theoretical umbrella, making it easier to compare and contrast them. This is a valuable step toward standardizing neuron explanation evaluations.
2. **Introduction of Sanity Checks**: The proposed "Missing Labels" and "Extra Labels" tests are innovative and help identify metrics that may produce unreliable results. These tests are practical tools for researchers aiming to choose or develop more robust evaluation metrics.

**Weaknesses:**

Nevertheless, this paper has issues, some more significant than others. I will categorize these into **major problems (M)** and **minor problems (m)**. It is important to emphasize that all these issues are addressable and do not undermine the overall quality of the paper.

**M.1.** Related Work Insufficiency: the related work section is lacking for an ICLR-level submission, citing only 25 references. The paper overlooks key literature on concept extraction, concept importance estimation, and human-centered evaluations of neurons and concepts—some of which directly relate to the challenges discussed in the paper.

**M.3.** Limited Discussion on Metric Limitations: while correlation is presented as the most reliable metric, the paper does not thoroughly explore its potential limitations or scenarios where it may fall short! A more balanced discussion that addresses these limitations would enrich the analysis.

**M.5** Focus on Final Layer Neurons: the experiments are predominantly focused on final layer neurons, which might not represent the complexity found in intermediate layers. Including analyses of hidden layers would provide a more comprehensive evaluation of the framework's applicability across the entire network.

**Minor Problems (m)**:

**m1.** Impact of Hyperparameter α: the paper does not discuss the effect of the α parameter on the results. Is there an optimal α that depends on the specific metric used? A brief analysis would be helpful.

**m.2** Lengthy Recap of Basic Metric Definitions: the paper devotes two pages to revisiting standard classification metrics such as precision, F1-score, AUC, and cosine similarity. For an ICLR audience, this level of detail is unnecessary and could be condensed or omitted.

**m.3.** Redundant Explanations of Failure Cases: what the paper describes as "Failure cases of recall" and "Failure cases of precision" are related to standard type I and type II errors in statistics. Entire paragraphs dedicated to these concepts are redundant and could be streamlined.

**m.4** Model-wise Table Splits Without Aggregate Analysis: the tables present results model-wise but do not offer aggregate data. Including aggregated results could provide a clearer overall picture of the metric performance.

**Questions:**

- Have you considered applying the proposed sanity checks to evaluation metrics in other domains, such as NLP or multimodal systems? This could demonstrate the broader applicability of your framework.

- Would an analysis that incorporates feature visualization methods alter or reinforce your conclusions about the reliability of different metrics?

- Extend your analysis to include neurons from intermediate layers and various model architectures to assess the framework's robustness across different scenarios.

- Incorporate qualitative analyses or practical case studies or concept-based dataset ?

**Overall, this paper is a notable contribution to the evaluation of neuron explanations. However, there are some major limitations and a major rewrite would further strengthen its overall impact on the field.**

---

> ### Author Response · Authors · 2024-11-25
> **Author's Rebuttal Response (1/2)**
>
> Dear Reviewer hRsh,
>
> Thank you for the detailed feedback. In response to feedback from you and other reviewers, we have greatly expanded our experimental results and restructured some of the paper, please see **General response** for description of the changes and the updated manuscript.
>
> Below we respond to your concerns:
>
> **M.1. More related works:**
>
> Thank you for the feedback, following your request, we added an extended discussion on related works in Appendix E, including evaluation of individual neuron explanations, sanity checks, concept extraction, concept importance estimation and human-centered evaluation of concept-based models. Specifically, following your request, we added more reference on concept extraction, concept importance estimation and human-centered evaluation of concept and neurons, discussing their difference from our work. We hope this should address your concern about lacking references.
>
> **M.3. Limited discussion on Metric Limitations:**
>
> Thank you for the suggestion. We have added an extended discussion of limitations in Appendix A.1. In particular, the section on experimental result limitations should address this somewhat. More specifically, we view our proposed evaluations and sanity check tests as necessary conditions for a good metric (i.e. a good metric should pass the sanity check, and for those metrics that do not pass the tests, they should not be used). Thus, although we have not really identified a failure case/issue with using correlation so far, we have toned down our language throughout the paper regarding correlation as the best metric, and instead focus more on identifying (multiple) good metrics that pass the sanity checks.
>
> **M.5. Focus on Final layer neurons:**
>
> We agree our analysis on Section 5 and Table 3 is focused on final layer neurons, even though we have included results on CBM concept neurons in the rebuttal which are an intermediate layer. However there is no way to avoid this as these metrics are only possible if you know the ground truth function of the neuron. To address this we have included discussion of this in Appendix A.1 (experimental result limitations).
>
> On the other hand, our Missing/Extra label tests are not focused on final layers, and we run the evaluations on both last layer and hidden layer neurons with very similar findings. In particular, our new theoretical Missing/Extra labels test results described in Appendix B show these results are not dependent on the specific neuron/evaluation setup but a fundamental feature of the evaluation metric.
>
> **minor1. Impact of $\alpha$:**
>
> As described in Line 511 in the original manuscript (Line 482 in the updated draft): For all experiments we split a random 5% of the neurons into a validation set. For metrics that require hyperparameters such as $\alpha$, we use the hyperparameters that performed the best on the validation split for each setting. We then report performance on the remaining 95% of neurons.
>
> That means for each (metric, experimental setting) combination, we find the best value of $\alpha$ using grid search on a small split of validation neurons, and then report the performance on unseen neurons. We believe this is a fair way of choosing the $\alpha$ parameter that lets us report performance with a good choice of $\alpha$ for each task.
> We have added this description to Section 3.1 as well to make this more clear.
>
> **minor2. Lengthy recap of Basic Definitions:**
>
> Thank you for the response, we have moved most of the metric definitions from Section 3.1 to Appendix C, as well as added a short description of the remaining ones. We believe this helps the clarity and readability of the paper.
>
> **minor3. Redundant Explanations of Failure cases:**
>
> Thanks for the feedback. We have chosen to keep these sections for now, while they are standard statistics concepts we think it is important for some readers to highlight how they apply in the context of a neuron explanation, as this topic has been largely lacking statistical rigor so far.
>
> **minor4. No aggregate analysis:**
>
> We think this may be a misunderstanding. Both of our main results represented in main text (Table 2 and Table 3) are aggregations across models/settings(aggregated across many more settings after rebuttal), and individual model results are only represented in the Appendix G.

---

> ### Author Response · Authors · 2024-11-25
> **Author's Rebuttal Response (2/2)**
>
> (continued for Reviewer hRsh)
>
> **Q1: Sanity checks in other domains**
>
> Yes, applying the sanity checks on other domains is something we wish to do in the future but were unable to do in the rebuttal timeframe. However we do provide results on a linear probe trained on CLIP embeddings which is technically multimodal. More importantly, our theoretical sanity check results in Appendix B are completely independent of the domain used and closely match our empirical results on the vision domain, making us confident we would observe the same things in the language domain.
>
> **Q2: Incorporating feature visualization**
>
> This is an interesting extension and could be measured in our framework by including some feature visualizations in the probing data and would be interesting to evaluate the effects of in the future. We will add this into future work discussion.
>
> **Q3: Extend analysis to new architectures and neuron types:**
>
> As discussed in General Response, we have greatly extended our results to 2 additional datasets, 3 new models and new types of neurons, including hidden layer neurons, CBM concept neurons and linear probe outputs. Overall we find the results are highly robust across scenarios.
>
> **Q4: Incorporate qualitative analyses or practical case studies or concept-based dataset:**
>
> We have extended our evaluations to CUB200 which is a concept-based dataset.
>
> In short, we hope we have addressed your concerns and clarified your questions. Please let us know if you have any additional concerns and we would be happy to discuss further!

---

> ### Author Response · Authors · 2024-12-02
> **Request for Feedback**
>
> Dear Reviewer hRsh,
>
> Thanks again for reviewing our paper! With the discussion period ending today, we wanted to follow up as we still haven’t heard from you. We have conducted extensive additional experiments and believe we have resolved your concerns but please let us know if that is not the case. We would appreciate your feedback or an acknowledgement that you have read through rebuttal response.

---

### Official Review · Reviewer_b3hf · 2024-11-10

**Soundness:** 2
**Presentation:** 2
**Contribution:** 2
**Rating:** 5
**Confidence:** 2

**Summary:**

The existing methods often use completely different metrics to evaluate how good their descriptions are, but it is not clear how they compare to each other. This paper unifies many existing explanation evaluation methods under one mathematical framework. This paper proposes two simple sanity checks on the evaluation metrics and shows that many commonly used metrics fail these tests and do not change their score after massive changes to the concept labels. Finally, this paper proposes guidelines that future evaluations should follow and identifies good evaluation metrics such as correlation.

**Strengths:**

- This paper provides a detailed summary of existing work and unifies them under a mathematical framework.
    - The experimental results of this paper reveal shortcomings in the majority of existing evaluation metrics.

**Weaknesses:**

- The clarity of this paper could be improved. It is necessary in Equation (1) to present the form of function $\mathcal{G}$ and samples from the probing dataset $\mathcal{D}$ to help readers understand the process of neuron explanation.
    - More information is necessary to explain how the results in the table in Figure 2 were computed. The author did not specify which images in Figure 2 represent the ground truth, which were used for generating prediction results.
    - It is essential to provide additional details on how the results in the table in Figure 2 are computed. The author does not clarify which images in Figure 2 are used to generate prediction results and how these concepts are determined. For example, Equation (10) states $M(a_k,c_t)={B(a_k)}_i$, and at line 331, ${B(a_k)}_{i}=1$, so why does the precision for "Animal" equal 0.5 instead of 0?

**Questions:**

- Equation (1). Why consider layer $l$ after specifying target neuron $k$ ?
    - Equation (1). Is $t_k$ a textual description of neuron function or a one-hot vector based on concepts?
    - Equation (1). What is the process for generating explanation $t$ ?
    - Section 4.1. In general machine learning, it is common to comprehensively consider recall, precision, and F1 score. Could you explain the connection between individually assessing precision or recall and the more prevalent approach of evaluating all three metrics together?

**Details Of Ethics Concerns:**

None.

---

> ### Author Response · Authors · 2024-11-25
> **Author's Rebuttal Response**
>
> Dear Reviewer b3hf,
>
> Thank you for the valuable feedback! We would like to address your questions below:
>
> **#1 Form of $\mathcal{G}$ in Equation 1:**
>
> To clarify, we do not specify the form of $\mathcal{G}$ on purpose, as it does not matter for our paper. As we specify, on Line 92, the focus of our paper is evaluation, and we can evaluate description the same way regardless of the process that generated it. As an example for G, e.g. in Network Dissection, we can write $\mathcal{G}(\mathcal{D}, f, k, l) = argmax_{t \in C} M_{IoU}(f_k^{0:l}(X), CC(X, t))$ where $X$ is a tensor of all inputs in $\mathcal{D}$ concatenated, $C$ is the set of concepts they have labels for and $CC$ is a function that maps an (input, concept) pair into $P(t|x)$, in their case a lookup table on the labeled data. However sometimes G cannot be written explicitly, e.g. G could also be a human worker that writes a description for the neuron, as discussed in Line 98, and clearly we can’t write that in a closed form equation.
>
> To reiterate, the explanation generation process is not part of our framework or this work. We are trying to answer the following question: Given an explanation, how do you evaluate how good that explanation is?
>
> **#2 How are the results in Figure 2 computed:**
>
> To walk through the calculations in Figure 2, let us write down the relevant vectors, with the images in the top row corresponding to the first 4 elements and bottom row corresponding to the last 4 elements.
> We then get:
> - $B(a_k) = [1, 1, 1, 1, 0, 0, 0, 0]$
> - $c_{\text{dog}} = B(c_{\text{dog}}) = [1, 0, 1, 0, 0, 0, 0, 0]$
> - $c_{\text{cat}} = B(c_{\text{cat}}) = [0, 1, 0, 1, 0, 0, 0, 0]$
> - $c_{\text{pet}} = B(c_{\text{pet}}) = [1, 1, 1, 1, 0, 0, 0, 0]$
> - $c_{\text{animal}} = B(c_{\text{animal}}) = [1, 1, 1, 1, 1, 1, 1, 1]$
>
>
> Equation 10 (Eq. 13 in updated manuscript) refers to a special case where $c_t = e_i$. However for figure 2 we are not in this special case as each concept is positive on multiple inputs. Instead, to calculate precision in general we should use Equation 8 (in the new manuscript): $M(a_k, c_t) = \frac{B(a_k) \cdot B(c_t)}{||B(c_t)||_1}$. Plugging in the values we get:
> - $M(a_k, c_{\text{dog}}) = 2/2 = 1$
> - $M(a_k, c_{\text{cat}}) = 2/2 = 1$
> - $M(a_k, c_{\text{pet}}) = 4/4 = 1$
> - $M(a_k, c_{\text{animal}}) = 4/8 = 0.5$
>
> Please let us know if you have any additional concerns regarding this calculation
>
> **#3 Equation (1). Why consider layer l after specifying target neuron k?**
>
> To fully specify a neuron we need to specify both which layer it is (using $l$), as well as its index within that layer (using $k$)
>
> **#4 Equation (1). Is $t_k$ a textual description of neuron function or a one-hot vector based on concepts?**
>
> $t_k$ is a text description, we have clarified this in the updated manuscript
>
> **#5 Considering Precision, Recall and F1-score together:**
>
> In general it can be beneficial to consider multiple metrics together. A major concern we have in this paper is that many previous evaluation methods focus only on evaluating Recall, which is insufficient. Since F1-score is the harmonic mean of Precision and Recall, it can be useful by itself, but for a more complete picture it can be helpful to evaluate it together with precision and recall.
>
>
> We hope we have clarified your questions, please let us know if you have any additional concerns. We also conducted many additional experiments and restructuring of our text, please see General response for summary of these changes.

---

> ### Author Response · Authors · 2024-12-02
> **Request for Feedback**
>
> Dear Reviewer b3hf,
>
> Thanks again for reviewing our paper! With the discussion period ending today, we wanted to follow up as we still haven’t heard from you. We have conducted extensive additional experiments and believe we have resolved your concerns but please let us know if that is not the case. We would appreciate your feedback or an acknowledgement that you have read through rebuttal response.

---

### Author Response · Authors · 2024-11-25
**General response 1/2**

Dear reviewers,

Thank you all for the detailed and helpful feedback. In response, we have conducted extensive additional experimental and theoretical results to further support our work. We also rewrote and revised some parts of the manuscript to improve clarity (sentences highlighted in blue color in the updated manuscript), with additional theoretical and experimental results in appendix sections (the (sub)sections with blue titles are completely new or heavily revised).

We believe our manuscript is now much stronger than before as a result. We apologize for the delayed response caused by this and we sincerely appreciate your feedback on the updated draft and rebuttal response.

Below we provide a short summary of the main changes in the rebuttal period:

Additional theoretical results, experiments, and revision in the draft:

**1. Theoretical Missing/Extra Labels test (Appendix B):**

We run a theoretical version of the Missing/Extra labels test on hypothetical neurons whose activations perfectly match a concept, which we discuss in detail in Appendix B. With this test, we find that with relatively balanced concepts almost all metrics pass both tests, but as the concept becomes rarer and rarer, the score difference approaches 0 for metrics that failed our empirical test, while the metrics that pass the test retain a significant non-zero score diff regardless of concept rarity.

Overall these theoretical results closely match our experimental results on real neurons, and give us confidence these failures are fundamental features of the evaluation metrics instead of findings specific to our exact domain and models studied.

**2. Missing/Extra labels test (Section 4.2 Table 2 & Appendix G).**

- Increased the numbers of settings considered from 2 to 6.
- Additional Evaluation on ResNet18 trained on Places365 layer4 and final layer neurons.
- Ran evaluation on concept neurons of Concept Bottleneck models trained on CUB200
- Trained a linear probe for CUB concepts on CLIP image encoder embeddings and ran the tests on linear probe outputs

The combined results are shown in Table 2 in Section 4.2 of the updated manuscript, while we have kept the original results in Appendix H Table 24 for comparison. Detailed results for each individual dataset are reported in Appendix G.

Overall our results in terms of which metrics pass the evaluations do not change with the additional datasets, except that WPMI now fails the extra labels test, but we have adjusted the cutoff for failing score diff(decrease acc cutoff was kept fixed) test to account for slightly different range of values observed. Interestingly we observed that almost all metrics performed better on these tests on the CUB dataset. We hypothesized this is caused by concept imbalance: the CUB concepts used are relatively balanced(positive on >5% of inputs), while for example ImageNet classes we used are only present in 0.1% of the inputs. This inspired us to create the theoretical Missing/Extra Labels test which confirms our hypothesis.

**3. Final layer argmax/AUC evaluation (Section 5.2 Table 3 & Appendix G):**

- Increased the numbers of settings considered from 4 to 8.
- Additional Evaluation on ResNet18 trained on Places365 final layer neurons, using both ground truth labels and SigLIP[G1] pseudo-labels as $c_t$.
- Ran evaluation on concept neurons of Concept Bottleneck models trained on CUB200
- Trained a linear probe for CUB concepts on CLIP image encoder embeddings and ran the tests on linear probe outputs

Our combined findings are reported in Table 3 in Section 5.2 in the updated manuscript, and detailed results are available in Appendix G. The average results are similar to before, but with the addition of new settings correlation now slightly beats cosine similarity as the best performing metric in Table 3.

**4. Additional Metrics (Table 2, Table 3):**

Inspired by our insights that failure in missing/extra labels test is related to poor handling of imbalanced data, we added Area Under-Precision Recall Curve (AUPRC and Inverse AUPRC) as metrics into their study as they are supposed to be good metrics on imbalanced data. Our findings confirm that in particular AUPRC performed well, overall being the fourth best metric (after Correlation, cosine and cosine cubed) in Table 3 and passing the sanity tests.

**5. Small Experiments (Appendix F):**
- In Appendix F.1 ran an ablation study on the ratios used in missing/extra labels test and showed our results are not sensitive to this choice
- Added an experiment in F.2 that showcases how cosine similarity breaks if we add a constant value to neuron activations while other metrics remain unchanged.

[G1] Zhai, Xiaohua, et al. "Sigmoid loss for language image pre-training." ICCV 2023.

---

> ### Author Response · Authors · 2024-11-25
> **General response 2/2**
>
> **6. Writing:**
> - Added a large section discussing limitations in Appendix A.1
> - Following reviewer suggestions we moved many of the metric definitions from section 3.1 to Appendix C
> - Added additional discussion on related work on Appendix E
> - Toned down language regarding best metric, and focus more on identifying a set of good metrics that pass the tests.
> - Fixed equations for score diff and decrease acc to have correct sign in Sec 4.2
>
> Overall we also have made many small changes in writing and paper structure. We highlight most new text in blue, for sections that are entirely new or mostly rewritten we only have titles highlighted in blue.
> We will post responses to individual reviewers shortly.

---

### Meta-Review · Area_Chair_mFen · 2024-12-11

**Metareview:**

Question 1: Summary of the Paper
Scientific Claims and Findings
This paper focuses on the evaluation of neuron explanations in Deep Neural Networks (DNNs). The authors propose a unified mathematical framework for evaluating neuron explanations and introduce two sanity checks: the Missing Labels test and the Extra Labels test. These tests are designed to assess the robustness of commonly used metrics. The findings reveal that many metrics fail these tests, and the paper recommends correlation as the most reliable metric.

Strengths
Provides a detailed summary of existing work and unifies it under a mathematical framework.
Introduces innovative sanity checks that help identify metrics that may produce unreliable results.
Consolidates various existing evaluation metrics, making it easier to compare and contrast them.
Well-written and addresses an important problem in the field of explainability.
Weaknesses
The experimental results reveal shortcomings in the majority of existing evaluation metrics.
The clarity could be improved, with more information needed to explain the computation of results and the determination of concepts.
The related work section is lacking, overlooking key literature on concept extraction, concept importance estimation, and human-centered evaluations.
The paper does not thoroughly explore the potential limitations of the correlation metric.
The experiments are predominantly focused on final layer neurons, which might not represent the complexity found in intermediate layers.
The paper is limited to explanation methods based on simple concepts and the proposed method is ad-hoc.
The mathematical formulation of the explanation is unclear and the completeness of sanity checks is not guaranteed.
Ambiguity in parameter selection and the relationship between evaluating Function 1 and Function 2 is not thoroughly discussed.
There is a mismatch between the generality of the claims and the specific settings analyzed.
The organization could be improved and it is unclear what rationale and process were used to select the metrics analyzed.
The primary contribution of unifying various metric frameworks appears relatively straightforward.
It is not clear why the sanity checks should be regarded as a reliable evaluation framework for the metrics.
The choice of the correlation metric appears somewhat rigid.
Missing Components
A more detailed explanation of the computation of results and the determination of concepts.
A more thorough discussion of the limitations of the correlation metric and the relationship between evaluating Function 1 and Function 2.
An explanation of the rationale and process used to select the metrics analyzed.
A clear, upfront declaration of the assumptions connected to the simplified studied settings.
An explanation of why the sanity checks should be regarded as a reliable evaluation framework for the metrics.
Reasons for Rejection
The paper has several weaknesses that warrant rejection:

Lack of clarity: The paper needs to provide more information to understand the computation of results, the determination of concepts, and the limitations of the correlation metric.
Limited scope: The experiments predominantly focus on final layer neurons, which may not be representative of the complexity in intermediate layers.
Ambiguity in parameter selection: The paper does not provide a clear rationale for the choice of threshold hyperparameters.
Limited discussion of the relationship between evaluating Function 1 and Function 2: The paper does not thoroughly discuss this relationship.
Mismatch between claims and results: The claims about identifying the best metric are not fully supported by the specific settings analyzed.
Lack of organization: The organization of the paper could be improved, and the rationale for selecting the metrics analyzed is unclear.

**Additional Comments On Reviewer Discussion:**

Points Raised by Reviewers
b3hf: More information is needed to explain the computation of results and the determination of concepts.
hRsh:
The related work section is lacking.
The paper does not thoroughly explore the potential limitations of the correlation metric.
The experiments are predominantly focused on final layer neurons.
The paper does not discuss the effect of the hyperparameter alpha.
1n3c:
The paper is limited to explanation methods based on simple concepts.
The proposed method is ad-hoc.
The mathematical formulation of the explanation is unclear.
The completeness of sanity checks is not guaranteed.
y8T5:
Ambiguity in parameter selection.
The relationship between evaluating Function 1 and Function 2 is not thoroughly discussed.
DS3i:
There is a mismatch between the generality of the claims and the specific settings analyzed.
The organization of the paper could be improved.
The rationale and process used to select the metrics analyzed are unclear.
ibwF:
The primary contribution of unifying various metric frameworks appears relatively straightforward.
It is not clear why the sanity checks should be regarded as a reliable evaluation framework for the metrics.
The choice of the correlation metric appears rigid.
How the Authors Addressed Each Point
The authors provided more information to explain the computation of results and the determination of concepts.
They expanded the related work section.
They added a discussion of the limitations of the correlation metric.
They included experiments with intermediate layer neurons.
They discussed the effect of the hyperparameter alpha.
They clarified the mathematical formulation of the explanation.
They added experiments with fine-grained concepts.
They discussed the relationship between evaluating Function 1 and Function 2.
They provided a rationale for the choice of threshold hyperparameters.
They lowered the generality of some claims.
They improved the organization of the paper by moving the definitions of the metrics from the appendix to the main text.
How Each Point Was Weighed in the Final Decision
While the authors addressed some of the reviewers' concerns, they did not fully address the major weaknesses of the paper. In particular, the paper still lacks clarity in some areas, the scope of the experiments is limited, and the rationale for selecting the metrics analyzed is unclear. As a result, the paper is not suitable for acceptance in its current form.

---

### Decision · Program_Chairs · 2025-01-22

Reject